# Solving the Offline and Online Min-Max Problem of Non-smooth Submodular-Concave Functions: A Zeroth-Order Approach

Amir Ali Farzin [1]   Yuen-Man Pun [1]   Philipp Braun [1]   Tyler Summers [2]   Iman Shames [3]

## Abstract

We consider max-min and min-max problems with objective functions that are possibly non-smooth, submodular with respect to the minimiser and concave with respect to the maximiser. We investigate the performance of a zeroth-order method applied to this problem. The method is based on the subgradient of the Lovász extension of the objective function with respect to the minimiser and based on Gaussian smoothing to estimate the smoothed function gradient with respect to the maximiser. In expectation sense, we prove the convergence of the algorithm to an $\epsilon$-saddle point in the offline case. Moreover, we show that, in the expectation sense, in the online setting, the algorithm achieves $O(\sqrt{N(1 + \bar{P}_N)})$ online duality gap, where $N$ is the number of iterations and $\bar{P}_N$ is the path length of the sequence of optimal decisions. The complexity analysis and hyperparameter selection are presented for all the cases. The theoretical results are illustrated via numerical examples.

## 1. Introduction

In this paper, we study mixed-integer max-min and min-max problems of the form

$$\max_{y \in \mathcal{Y}} \min_{S \subset Q} f(S, y) \quad \text{and} \quad \min_{S \subset Q} \max_{y \in \mathcal{Y}} f(S, y). \quad (1)$$

Here, the objective function $f \colon 2^Q \times \mathbb{R}^m \to \mathbb{R}$ is a set function over a ground set $Q$ of $n \in \mathbb{N}$ elements, where $2^Q$ denotes the power set of $Q$, $f(S, \cdot)$ is a continuous possibly

[1]School of Engineering, Australian National University, Canberra, Australia [2]Department of Mechanical Engineering, University of Texas at Dallas, Richardson, Texas, USA [3]Department of Electrical and Electronic Engineering, The University of Melbourne, Parkville, Australia. Correspondence to: Amir Ali Farzin <amirali.farzin@anu.edu.au>.

*Proceedings of the 43rd International Conference on Machine Learning*, Seoul, South Korea. PMLR 306, 2026. Copyright 2026 by the author(s).

non-differentiable function with respect to the second variable $y \in \mathcal{Y}$ and $\mathcal{Y} \subset \mathbb{R}^m$ is a compact convex constraint set with diameter $D_y$. Without loss of generality, we assume $Q = [n] = \{1, 2, \ldots, n\}$. Moreover, we assume a saddle point for $f$ exists (which exists when solutions to (1) coincide), and consider computing an $\epsilon$-saddle point, i.e., $\bar{S} \in 2^{[n]}, \bar{y} \in \mathcal{Y}$ with $\max_{y \in \mathcal{Y}} f(\bar{S}, y) - \min_{S \in 2^{[n]}} f(S, \bar{y}) \le \epsilon$, and whose definition and existence are made precise in the main part of the paper.

In this work, we specifically examine max-min and min-max optimisation problems whose objective functions exhibit a submodular-concave structure. An example of the applications of such a problem is the offline or online adversarial image segmentation or adversarial semi-supervised clustering. To the best of our knowledge, this is the first study on solving min-max optimisation problems with a non-smooth submodular-concave cost function. Since the foundational contribution of (Edmonds, 1970), submodular minimisation has become a fundamental tool across numerous domains. Its influence spans machine learning (Krause & Cevher, 2010), economics (Topkis, 1998), computer vision (Jegelka et al., 2013), and speech recognition (Lin & Bilmes, 2011). In many applications, it is crucial to obtain solutions that remain reliable in the presence of noise, outliers, or adversarial perturbations. Optimisation problems that seek robustness against the worst-case scenario are commonly formulated as min-max problems. Consequently, the past several years have seen significant progress on continuous min-max optimisation (Daskalakis et al., 2018; Farzin et al., 2025a; Mokhtari et al., 2020; Farzin et al., 2025b). This research direction has produced a diverse collection of algorithms and a steadily expanding range of applications, including adversarial example generation (Wang et al., 2021), robust statistics (Agarwal & Zhang, 2022), and multi-agent systems (Li et al., 2019).

To guarantee the existence and computability of saddle points, certain structural assumptions are typically required. The rich structure of submodularity enables us to exploit combinatorial properties when analysing and solving these min-max problems. Previous studies that analysed leveraging submodularity in offline min–max problems include (Adibi et al., 2022). Here, the authors considered offline min-

max problems with convex-submodular objective functions. They showed that obtaining saddle points is hard up to any approximation for that class of problems, and introduced new notions of near-optimality. Based on these notions, the authors provide algorithmic procedures for solving convex-submodular min-max problems and characterise their convergence rates and computational complexity. Since minimisation and maximisation of submodular functions are fundamentally different problems (Bach, 2013), a direct comparison of the results of this work and (Adibi et al., 2022) is not sensible. In (Mualem et al., 2024), the authors considered submodular offline integer min-max and offline integer max-min problems and they have established a systematic study of min-max optimisation for combinatorial problems. They have mapped the theoretical approximability of max-min submodular optimisation, and also obtained some understanding of the approximability of min-max submodular optimisation. In this work, we are going to consider the case of both offline and online submodular-concave non-smooth mixed-integer max-min and min-max problems to fill this gap in the literature of combinatorial min-max problems.

A central observation we will rely on is that minimising a submodular set function (SFM) is equivalent to minimising its Lovász extension over the hypercube $[0,1]^n$ (Lovász, 1983). The Lovász extension possesses several favourable analytic properties, most notably convexity and Lipschitz continuity, which make it a powerful tool for addressing optimisation problems. However, the Lovász extension is inherently non-smooth, meaning that standard gradient-based methods cannot be applied directly to locate its minimisers. As a result, algorithms for both offline SFM (Hamoudi et al., 2019; Axelrod et al., 2020) and online SFM (Hazan & Kale, 2012; Jegelka & Bilmes, 2011; Cardoso & Cummings, 2019) typically rely on subgradients of the Lovász extension. In (Hazan & Kale, 2012) it is shown that the subgradient of the Lovász extension of a submodular function can be obtained only by function evaluations and no higher order information is needed to calculate the subgradient. Given this non-smooth structure of Lovász extension and our setting where the objective function is possibly non-smooth with respect to the maximiser, zeroth-order optimisation methods naturally emerge as an appropriate class of techniques for solving (1). Optimisation frameworks that rely only on the evaluations of the objective function (but not on its derivatives) are commonly called zeroth-order (ZO) or derivative-free methods (Audet & Hare, 2017). Multiple ZO algorithms have been developed to solve various optimisation problems. The authors in (Nesterov & Spokoiny, 2017), proposed and analysed a method for minimisation problems based on ZO oracles leveraging Gaussian smoothing. This method has been extensively studied for minimisation problems with different classes of functions in both

offline (Farzin & Shames, 2024; Farzin et al., 2025d) and online settings (Shames et al., 2019; Farzin et al., 2025c). Moreover, the performance of ZO algorithms leveraging random Gaussian oracles in different continuous (non)convex-(non)concave (non)smooth conditions has been analysed previously (Wang et al., 2023; Farzin et al., 2025a). Besides the ZO methods, in (Nedić & Ozdaglar, 2009), the authors considered solving non-smooth convex-concave min-max problems using a first-order primal-dual subgradient method, and they show that the averaged iterates yield an $\epsilon$-approximate saddle point in $O(\epsilon^{-2})$ iterations.

In this work, we develop a ZO framework built on the extragradient method to solve Problem 1 in both offline and online regimes. Our approach requires only function evaluations, and no gradient or higher-order information is needed. We first analyse the offline setting and show that the proposed ZO method converges, in expectation, to an $\epsilon$-saddle point using $O(nm^2\epsilon^{-2})$ function queries. We then turn to the online setting and demonstrate that the method achieves an expected online duality gap of order $O(\sqrt{N\bar{P}_N})$, where $N$ denotes the number of iterations and $\bar{P}_N$ is the path length of the sequence of optimal decisions. This dependence on path length is consistent with the well-established behaviour of dynamic regret bounds in online optimisation (Zinkevich, 2003; Hazan, 2016). Finally, we illustrate the theoretical findings through several numerical examples. We show that the proposed method can outperform supervised and semi-supervised trained U-Net (introduced in (Ronneberger et al., 2015)) in online adversarial image segmentation.

*Outline:* Section 2 presents the necessary preliminaries. Section 3 introduces our main algorithmic framework and establishes its convergence and complexity guarantees. An illustrative example is provided in Section 4. Finally, Section 5 concludes the paper and outlines several promising directions for future research. Background material, additional examples and proofs of the main results are collected in Appendices A-E.

*Notation:* For $x, y \in \mathbb{R}^d$, $\langle x, y \rangle = x^\top y$ and $\|\cdot\|$ denotes the Euclidean norm of its argument. The gradient of a differentiable function $f : \mathbb{R}^d \to \mathbb{R}$ is denoted by $\nabla f$. The ceiling of a real number $N \in \mathbb{R}$, i.e., the smallest integer greater than or equal to $N$, is denoted by $\lceil N \rceil$. Identity matrix of proper dimension is denoted by $\mathbb{I}$. Let $\{0,1\}^n \subset \mathbb{N}^n$ denote the set of $n$-dimensional binary vectors, where each coordinate is either 0 or 1. There is a natural bijection between a vector $x \in \{0,1\}^n$ and a subset $S \subset [n] := \{1, 2, \ldots, n\}$. For a set $S \subset [n]$, let $\chi_S \in \{0,1\}^n$ denote its characteristic vector, defined by $\chi_S(i) = 1$ if $i \in S$ and $\chi_S(i) = 0$ otherwise. We will freely move between these two representations and, in particular, interchangeably use

$$f(S), \ S \subset [n] \quad \text{and} \quad f(\chi_S), \ \chi_S \in \{0,1\}^n. \quad (2)$$

We denote the cardinality of the set $S$ with $|S|$. The function $F : D \to \mathbb{R}$ with $D \subset \mathbb{R}^n$ is Lipschitz if there exists a *Lipschitz constant* $L_0 > 0$, such that

$$|F(x) - F(y)| \leq L_0 \|x - y\|, \quad \forall\, x, y \in D. \tag{3}$$

## 2. Preliminaries

In this section, we present the necessary background material of min-max problems, submodular functions, the Lovász extension and Gaussian smoothing. In addition, we make the problem formulation and the solution approach to solve (1) more precise. We assume that $Q = [n]$, $\mathcal{Y} \subset \mathbb{R}^m$ is convex and compact, and $f$ is continuous with respect to the maximiser, which will be used in derivations and definitions in this and in the following sections.

### 2.1. Min-Max Problems and Saddle Points

For the max-min/min-max problems in (1), we aim to determine a point that is a saddle point of the objective function, which is defined below, following (Adibi et al., 2022).

**Definition 2.1.** A pair $(S^*, y^*) \in 2^{[n]} \times \mathcal{Y}$, $n \in \mathbb{N}$, $\mathcal{Y} \subset \mathbb{R}^m$, is called a saddle point of the function $f : 2^{[n]} \times \mathcal{Y} \to \mathbb{R}$ if it satisfies the following condition:

$$\forall S \subset [n],\ y \in \mathcal{Y}: \quad f(S^*, y) \leq f(S^*, y^*) \leq f(S, y^*).$$

Based on this definition, a pair $(S^*, y^*)$ is a saddle point of the max-min Problem (1) if neither player has an incentive to unilaterally deviate from their respective strategies: when the maximisation variable is fixed at $S^*$, the minimisation variable $y^*$ remains optimal, and vice versa. In this sense, a saddle point represents an equilibrium of the minimax problem. There exists an extensive body of work on algorithms designed to efficiently find an $\epsilon$-saddle point in convex-concave minimax optimisation, where $\epsilon > 0$ is an arbitrary tolerance (Thekumparampil et al., 2019). To define an $\epsilon$-saddle point, we introduce the duality gap

$$D(S, y) := \bar{\phi}(S) - \underline{\phi}(y)$$

$$\text{where} \quad \bar{\phi}(S) := \max_{y \in \mathcal{Y}} f(S, y), \quad \underline{\phi}(y) := \min_{S \subset [n]} f(S, y).$$

**Definition 2.2.** A pair $(\bar{S}, \bar{y}) \in 2^{[n]} \times \mathcal{Y}$, $n \in \mathbb{N}$, $\mathcal{Y} \subset \mathbb{R}^m$, is an $\varepsilon$-saddle point of $f : 2^{[n]} \times \mathcal{Y} \to \mathbb{R}$ if it satisfies

$$D(\bar{S}, \bar{y}) = \bar{\phi}(\bar{S}) - \underline{\phi}(\bar{y}) \leq \varepsilon.$$

Definitions 2.1 and 2.2 coincide when $\epsilon = 0$. Consequently, finite-time analyses typically focus on finding an approximate $\varepsilon$-saddle point. Moreover, for a function $f$ with saddle point $(S^*, y^*)$, we can define the restricted gap as

$$R(\bar{S}, \bar{y}) = f(\bar{S}, y^*) - f(S^*, \bar{y}). \tag{4}$$

By adding and subtracting $f(S^*, y^*)$ to the right-hand side of the equation and from Definition 2.1, one can see that $R(\bar{S}, \bar{y}) \geq 0$ for all $S \subset [n]$ and $y \in \mathcal{Y}$. Moreover, noting

$$f(S^*, y^*) \leq f(\bar{S}, y^*) \leq \max_{y \in \mathcal{Y}} f(\bar{S}, y)$$

$$\min_{S \subset [n]} f(S, \bar{y}) \leq f(S^*, \bar{y}) \leq f(S^*, y^*),$$

it follows that $0 \leq R(\bar{S}, \bar{y}) \leq D(\bar{S}, \bar{y})$. In this work, we aim to find an $\epsilon$-saddle point of (1) by leveraging the convexity of the Lovász extension of a submodular function.

### 2.2. Submodular Functions and the Lovász Extension

In this work, we consider Problems (1) with submodular-concave cost functions. In what follows, we provide the background pertinent to minimising submodular functions and their Lovász extensions.

**Definition 2.3** (Submodular Functions (Hazan & Kale, 2012), Sec 2.1). A function $\bar{f} : 2^{[n]} \to \mathbb{R}$ is called submodular if it exhibits diminishing marginal returns. Formally, for all $S \subset T \subset [n]$ and any element $i \notin T$,

$$\bar{f}(S \cup \{i\}) - \bar{f}(S) \geq \bar{f}(T \cup \{i\}) - \bar{f}(T).$$

An equivalent characterisation is that for all $S, T \subset [n]$,

$$\bar{f}(S) + \bar{f}(T) \geq \bar{f}(S \cap T) + \bar{f}(S \cup T).$$

The well-known Lovász extension provides a continuous, convex extension of a submodular function $\bar{f} : 2^{[n]} \to \mathbb{R}$ to a function $\bar{f}^L : [0, 1]^n \to \mathbb{R}$, where $[0, 1]^n \subset \mathbb{R}^n$ is the unit hypercube. We now recall its definition.

**Definition 2.4** (Lovász Extension (Hazan & Kale, 2012), Def 5). Let $x \in [0, 1]^n$ and let $A_i \subset [n]$ $(i \in [n], n \in \mathbb{N})$, be a chain of subsets $A_0 \subset A_1 \subset \cdots \subset A_p$, $p \leq n$, such that $x$ can be expressed as a convex combination $x = \sum_{i=0}^{p} \lambda_i \chi_{A_i}$ where $\lambda_i > 0$ and $\sum_{i=0}^{p} \lambda_i = 1$. Then the value of the Lovász extension $\bar{f}^L$ at $x$ is defined to be

$$\bar{f}^L(x) = \sum_{i=0}^{p} \lambda_i \bar{f}(A_i). \tag{5}$$

*Remark* 2.5 ((Hazan & Kale, 2012), Def. 5). Sampling a threshold $\tau \in [0, 1]$ uniformly at random, and considering the level set $S_\tau = \{i \in [n] : x(i) > \tau\}$, we have

$$\bar{f}^L(x) = E_\tau[\bar{f}(S_\tau)]. \tag{6}$$

Remark 2.5, implies that for all sets $S \subset [n]$, it follows that $\bar{f}^L(\chi_S) = \bar{f}(S)$. Next, we state the main assumption on the structure of $f$ and prove a lemma that plays a crucial role in proving the main results of this paper.

**Assumption 2.6** (Submodular-Concave Structure). Function $f : 2^{[n]} \times \mathbb{R}^m \to \mathbb{R}$ is submodular with respect to its first variable and concave with respect to its second variable.

**Lemma 2.7.** *Under Assumption 2.6, let* $f^L : [0,1]^n \times \mathbb{R}^m \to \mathbb{R}$ *be the Lovász extension of* $f$ *with respect to its first variable. Then,* $f^L$ *is a convex-concave function. Moreover,* $\max_{y \in \mathcal{Y}} f^L(x,y) = \max_{y \in \mathcal{Y}} E_\tau[f(S_\tau, y)]$, *where* $S_\tau = \{i \in [n] : x(i) > \tau\}$.

A proof of Lemma 2.7 is given in Appendix C. In (Axelrod et al., 2020, Lem. 2.2), the authors showed that we can go from an approximate minimiser of $f^L$ to an approximate minimiser of $f$ in $O(n)$ function calls. Thus, leveraging this fact and considering Lemmas A.1 and 2.7, we can consider the Lovász extension of $f(S,y)$ with respect to the first variable, solve the non-smooth convex-concave min-max problem of the form

$$\min_{x \in [0,1]^n} \max_{y \in \mathcal{Y}} f^L(x,y), \tag{7}$$

instead of Problem (1), and recover the solution of the min-max Problem (1). As $f^L$ is non-smooth, ZO methods are good candidates to solve (7). Next, we define the maximal chain associated with the vector $x$ and the subgradient of the Lovász extension of a submodular function.

**Definition 2.8** (Maximal Chain Associated with $x$ (Hazan & Kale, 2012), Def. 6). Let $x \in [0,1]^n$, and let $A_0 \subset A_1 \subset \cdots \subset A_p$, $p \leq n$, be the unique chain such that $x = \sum_{i=0}^p \mu_i \chi_{A_i}$, where $\mu_i > 0$ and $\sum_{i=0}^p \mu_i = 1$. A maximal chain associated with $x$ is any maximal completion of the $A_i$ chain; that is, a maximal chain

$$\emptyset = B_0 \subset B_1 \subset B_2 \subset \cdots \subset B_n = [n]$$

such that for all $i \in [p]$ there exists $j \in [n]$ with $A_i = B_j$.

**Definition 2.9** (Lovász Extension Subgradient (Hazan & Kale, 2012), Prop. 7). Let $\bar{f} : 2^{[n]} \to \mathbb{R}$ be a submodular function and $\bar{f}^L$ be its Lovász extension. Let $x \in [0,1]^n$, let $\emptyset = B_0 \subset B_1 \subset B_2 \subset \cdots \subset B_n = [n]$ be an arbitrary maximal chain associated with $x$, and let $\pi : [n] \to [n]$ be the corresponding permutation. Then, a subgradient $g_x = [g_1, \ldots, g_n]^\top$ of $\bar{f}^L$ at $x$ is given by

$$g_i = \bar{f}(B_{\pi(i)}) - \bar{f}(B_{\pi(i)-1}), \qquad i \in [n]. \tag{8}$$

Complementary definitions, and lemmas related to submodular functions are given in Appendix A.

### 2.3. Relations between (1) and (7)

In this section, we discuss sufficient conditions for the existence of saddle points $(S^*, y^*)$ for $f$. Moreover, we explain the relationship between the solutions to (1) and (7).

**Proposition 2.10.** *Let* $f : 2^{[n]} \times \mathbb{R}^m \to \mathbb{R}$ *be a submodular-concave function and* $f^L$ *be the Lovász extension of* $f$ *with respect to the first variable. In general, the existence of a saddle point for* $f$ *is not guaranteed, and we have*

$$\max_{y \in \mathcal{Y}} \min_{S \subset [n]} f(S,y) \leq \min_{S \subset [n]} \max_{y \in \mathcal{Y}} f(S,y). \tag{9}$$

*The existence of a saddle point for* $f$ *is guaranteed and solutions of* (1) *and* (7) *coincide if*

$$\min_{x \in [0,1]^n} \max_{y \in \mathcal{Y}} f^L(x,y) = \min_{x \in \{0,1\}^n} \max_{y \in \mathcal{Y}} f^L(x,y). \tag{10}$$

A proof of Proposition 2.10 is given in Appendix C. Next, we give some conditions that guarantee that (10) holds.

**Proposition 2.11.** *Let* $f : 2^{[n]} \times \mathbb{R}^m \to \mathbb{R}$ *be a submodular-concave function and* $f^L$ *be the Lovász extension of* $f$ *with respect to the first variable. Let* $\zeta(x) = \max_{y \in \mathcal{Y}} f^L(x,y)$ *for* $\mathcal{Y} \subset \mathbb{R}^m$ *convex and compact. Then,* (10) *holds, and the existence of a saddle point for* $f$ *is guaranteed if one of the following conditions is satisfied: i)* $\zeta(x)$ *itself is a Lovász extension of a submodular function. ii) The set of minimisers of* $\zeta(x)$ *contains a vertex. iii) The family of functions* $\{f^L(\cdot, y) : y \in \mathcal{Y}\}$ *has a common vertex minimiser.*

A proof of Proposition 2.11 is given in Appendix C. In Appendix B, we present two simple examples for the case when $\max_{y \in \mathcal{Y}} \min_{S \subset [n]} f(S,y) < \min_{S \subset [n]} \max_{y \in \mathcal{Y}} f(S,y)$, i.e., $f(S,y)$ has no saddle point and for the case that $\max_{y \in \mathcal{Y}} \min_{S \subset [n]} f(S,y) = \min_{S \subset [n]} \max_{y \in \mathcal{Y}} f(S,y)$, i.e., $f(S,y)$ has a saddle point. Throughout the following sections, we assume that a saddle point (can be non-unique) for problems (1) exists. Thus, we can focus on solving (7) instead of (1) and leverage Lemma 2.7. As $f^L(x,y)$ is non-differentiable with respect to $x$, and we consider the case that it is possibly non-smooth with respect to $y$, we use ZO algorithms to solve (7). In the next sections, we introduce a ZO framework for solving (7) and investigate its performance properties.

### 2.4. Gaussian Smoothing

Following (Nesterov & Spokoiny, 2017), we define the Gaussian smoothed version of a continuous function $\tilde{f} : \mathbb{R}^m \to \mathbb{R}$, termed $\tilde{f}_\mu$ as

$$\tilde{f}_\mu(y) = \frac{1}{\kappa} \int_{\mathbb{R}^m} \tilde{f}(y + \mu u) e^{-\frac{1}{2} u^\top B u} \mathrm{d}u,$$
$$\kappa = \int_{\mathbb{R}^m} e^{-\frac{1}{2} u^\top B u} \mathrm{d}u = \frac{(2\pi)^{m/2}}{[\det B]^{\frac{1}{2}}}, \tag{11}$$

where vector $u \in \mathbb{R}^m$ is sampled from $\mathcal{N}(0, B^{-1})$, $B \in \mathbb{R}^{m \times m}$ is positive definite, and $\mu \in \mathbb{R}_{>0}$ is the smoothing parameter. In this work, we assume $B = \mathbb{I}$. In (Nesterov & Spokoiny, 2017), it is shown that independent of $\tilde{f}$ being differentiable or not, $\tilde{f}_\mu$ is always differentiable, and $\nabla \tilde{f}_\mu(y) = E_u[\frac{\tilde{f}(y+\mu u)}{\mu} u]$. Instead of this one-point gradient estimation, to reduce the variance of the estimator, we consider the two-point estimator and define the random oracle $g_\mu$ as

$$g_\mu(y) = \frac{1}{\mu}(\tilde{f}(y + \mu u) - \tilde{f}(y)) B u. \tag{12}$$

# 3. Main Results

Here, we first introduce the proposed ZO framework for solving (7) from which a solution for (1) can be deduced. Next, we investigate the application of the ZO algorithm in solving offline and online min-max problems with possibly non-differentiable submodular-concave cost functions.

## 3.1. Zeroth-order Framework

Here, we introduce an algorithm to solve Problem 1. Let

$$z = \begin{bmatrix} x \\ y \end{bmatrix}, \; G(z) = \begin{bmatrix} g_x(z) \\ -g_{\mu_y}(z) \end{bmatrix}, \; \mathcal{Z} = [0,1]^n \times \mathcal{Y}, \; (13)$$

where $g_x$ is the subgradient of $f^L$ with respect to $x$ defined in Definition 2.9 and $g_{\mu_y}$ is the random oracle obtained similar to (12) using $f^L$ as below

$$g_{\mu_y}(y) = \tfrac{1}{\mu}(f^L(x, y + \mu u) - f^L(x, y))u, \quad (14)$$

where $u \sim \mathcal{N}(0, \mathbb{I})$. Moreover, let

$$\hat{F}(z) = \begin{bmatrix} g_x(z) \\ -\nabla_y f_\mu^L(z) \end{bmatrix}, \; F(z) = \begin{bmatrix} g_x(z) \\ -g_y(z) \end{bmatrix}, \quad (15)$$

where $\nabla_y f_\mu^L$ is the gradient of the Gaussian smoothed version of $f^L$ with respect to $y$ and $g_y$ is the supergradient of $f^L$ with respect to $y$ (or $-g_y$ is subgradient of $-f^L$ with respect to $y$). It is straightforward to see that $E_u[G(z)] = \hat{F}(z)$ (Nesterov & Spokoiny, 2017), (21).

As $f^L(x, y)$ is non-differentiable with respect to $x$ and possibly non-differentiable in $y$, ZO methods and subgradient methods are natural selections to solve (7). We assume that access to $f$ is provided only through a value oracle. That is, given any set $S \subset [n]$ and $y \in \mathcal{Y}$, the oracle returns the value $f(S, y)$. Considering Definition 2.9, we can calculate $g_x$ simply by queries of $f$ without the need of extra information. We assume there is no access to the supergradient of $f^L$ with respect to $y$, which is a first-order information, and instead we use the random oracle (14) to update $y$. Hence, calculating $G(z)$ in (13) only requires the zeroth-order information. The overall algorithm to solve (7) is summarised in Algorithm 1.

In each iteration $k \in [N]$ of Algorithm 1, first, we sample $t_k \in \mathbb{N}$ number of directions from the standard Gaussian distribution and calculate the random oracles and leverage the average of the oracles. It is trivial to see that with $t_k$ samples instead of one, still $E[g_\mu(x_k)] = \nabla f_\mu^L(x_k)$, while the variance of $g_\mu(x_k)$ will be reduced (Farzin et al., 2025a). Then, we calculate the subgradient $g_x$ using (8) and obtain $G(z_k)$ defined in (13). By performing a descent step from $z_k$ using $G(z_k)$ with $h_{1_k}$ as the step size and projecting back to $\mathcal{Z}$, we obtain $\hat{z}_k$. Similarly, we obtain $G(\hat{z}_k)$ and perform a descent step from $z_k$ using $G(\hat{z}_k)$ with

---

**Algorithm 1** ZO-EG

1: **Input:** $z_0 \in \mathcal{Z}$, $\{h_{1_k}\}_{k=0}^N \subset \mathbb{R}_{>0}, \{h_{2_k}\}_{k=0}^N \subset \mathbb{R}_{>0}$, $\mu > 0$, $N, t_k \in \mathbb{N}$
2: **for** $k = 0, \dots, N$ **do**
3:    // (i) Forward step: estimate the operator at $z_k$
4:    Sample $u_k^1, \dots, u_k^{t_k}$ from $\mathcal{N}(0, \mathbb{I})$
5:    Compute $g_{\mu_y}^{1:t_k}(z_k)$ using $u_k^{1:t_k}$ and (14).
6:    Obtain $g_{\mu_y}(z_k) = \frac{1}{t_k} \sum_{i=1}^{t_k} g_{\mu_y}^i(z_k)$
7:    Calculate $g_x(z_k)$ using (8)
8:    Generate $G(z_k)$ using (13)
9:    // (ii) Extrapolation step: lookahead and re-estimate
10:    $\hat{z}_k = \text{Proj}_{\mathcal{Z}}(z_k - h_{1_k} G(z_k))$
11:    Sample $\hat{u}_k^1, \dots, \hat{u}_k^{t_k}$ from $\mathcal{N}(0, \mathbb{I})$
12:    Compute $g_{\mu_y}^{1:t_k}(\hat{z}_k)$ using $\hat{u}_k^{1:t_k}$ and (14).
13:    Obtain $g_{\mu_y}(\hat{z}_k) = \frac{1}{t_k} \sum_{i=1}^{t_k} g_{\mu_y}^i(\hat{z}_k)$
14:    Calculate $g_x(\hat{z}_k)$ using (8)
15:    Generate $G(\hat{z}_k)$ using (13)
16:    // (iii) Projection and update
17:    $z_{k+1} = \text{Proj}_{\mathcal{Z}}(z_k - h_{2_k} G(\hat{z}_k))$
18: **end for**
19: return $\hat{z}_k$ and $z_k$ for all $k = 0, \dots, N$.

---

$h_{2_k}$ as the step size and project back to $\mathcal{Z}$ to obtain $z_{k+1}$. The projection operator $\text{Proj}_{\mathcal{Z}} : \mathbb{R}^{n+m} \to \mathcal{Z}$ is defined as $\text{Proj}_{\mathcal{Z}}(\bar{z}) = \arg\min_{z \in \mathcal{Z}} \|z - \bar{z}\|^2$. Algorithm 1 returns the sequences $\hat{z}_k$ and $z_k$, which we can guarantee that they include an $\epsilon$-saddle point of $f^L$. A more detailed discussion is given in Theorem 3.2.

We know that $f^L$ is continuous and piecewise affine with respect to $x$, and for fixed $y \in \mathcal{Y}$, $f^L(\cdot, y)$ is Lipschitz continuous with some Lipschitz constant $L_{0x} > 0$ defined through the maximal slope of the piecewise affine function. Since $\mathcal{Y}$ is compact and considering (Rockafellar, 1997, Thm. 24.7), the subgradient of $f^L(\cdot, y)$ is upper bounded by $L_{0x}$, i.e., $\|g_x(x, y)\| \leq L_{0x}$ for all $x \in [0, 1]^n$, for all $y \in \mathcal{Y}$. Moreover, based on $f^L$ being Lipschitz, we have $|f^L(x, y)| \leq M$ for all $x \in [0, 1]^n$ and $y \in \mathcal{Y}$, where $M$ is a positive scalar. Following (Hazan & Kale, 2012), in each iteration of Algorithm 1 we choose thresholds $\tau, \hat{\tau} \in [0, 1]$ uniformly at random, and define the sets

$$S_k = \{i \in [n] : x_k(i) > \tau\}, \quad \hat{S}_k = \{i \in [n] : \hat{x}_k(i) > \hat{\tau}\}. \tag{16}$$

Next, we present our main results. We conclude this section with a general assumption used to simplify the statements.

**Assumption 3.1.** Function $f : 2^{[n]} \times \mathbb{R}^m \to \mathbb{R}$ is Lipschitz continuous with respect to the second variable with Lipschitz constant $L_{0y} > 0$.

### 3.2. Offline Settings

In this section, we consider the offline version of Problem (1) and investigate the performance of Algorithm 1 in finding an $\epsilon$-saddle point (according to Definition 2.2) of the cost function. The first theorem shows that for proper choices of the hyperparameters, the sequence generated by Algorithm 1 converge to an $\epsilon$-saddle point of $f^L$ and $f$, in expectation.

**Theorem 3.2.** *Consider Algorithm 1 with output $\{z_k\}_{k \geq 0}$ and $\{\hat{z}_k\}_{k \geq 0}$ and let $f : 2^{[n]} \times \mathbb{R}^m \to \mathbb{R}$ satisfy Assumptions 2.6 and 3.1. Let $f^L$ be the corresponding Lovász extension with respect to $x$ with $z^* = (x^*, y^*) \in [0,1]^n \times \mathcal{Y}$ as a saddle point. Let $L_0 = \min\{L_{0x}, 4M\}$, $N \geq 0$ be the number of iterations, $t_k = t = 1$ be the number of samples, $\mu > 0$ be the smoothing parameter in (14), and $\mathcal{U}_k = [u_0, \hat{u}_0, u_1, \hat{u}_1, \cdots, u_k, \hat{u}_k]$. Then, for any iteration $N$, with constant step sizes $h_{1,k} = h_1$ and $h_{2,k} = h_2$, we have the restricted gap of $f^L$ at $\hat{z}_k$ bounded by*

$$\frac{1}{N+1} \sum_{k=0}^{N} E_{\mathcal{U}_k}[R^L(\hat{z}_k)] \leq \frac{r_0^2}{2h_2(N+1)} \tag{17}$$
$$+ \left(\frac{h_2}{2} + h_1\right)(L_0^2 + L_{0y}^2(m+4)^2) + \mu L_{0y} m^{1/2},$$

*where $r_0 = \|z_0 - z^*\|$ and $R^L(\bar{z}) = f^L(\bar{x}, y^*) - f^L(x^*, \bar{y})$. Additionally, let $z_N^{\text{av}} = (x_N^{\text{av}}, y_N^{\text{av}}) = \frac{1}{N+1}\sum_{k=0}^{N} \hat{z}_k$ denote the averaged iterate. Then, the duality gap of $f^L$ at $z_N^{\text{av}}$ is bounded by*

$$E_{\mathcal{U}_N}[D^L(z_N^{\text{av}})] \leq \frac{\bar{r}_0^2}{h_2(N+1)} \tag{18}$$
$$+ (h_2 + h_1)(L_0^2 + L_{0y}^2(m+4)^2) + \mu L_{0y} m^{1/2},$$

*where $\bar{r}_0 = \max_{z \in \mathcal{Z}} \|z_0 - z\|$ and $D^L(\bar{z}) = \max_{y \in \mathcal{Y}} f^L(\bar{x}, y) - \min_{x \in [0,1]^n} f^L(x, \bar{y})$. Moreover, let $\epsilon > 0$, $\mu \leq \frac{\epsilon}{2L_{0y}m^{\frac{1}{2}}}$,*

$$h_1 \leq \frac{1}{(N+1)^{\frac{1}{2}}(L_0^2 + L_{0y}^2(m+4)^2)^{1/2}},$$
$$h_2 = \frac{\bar{r}_0}{(N+1)^{\frac{1}{2}}(L_0^2 + L_{0y}^2(m+4)^2)^{1/2}},$$
$$N \geq \left\lceil \frac{4(2\bar{r}_0+1)^2(L_0^2 + L_{0y}^2(m+4)^2)}{\epsilon^2} - 1 \right\rceil.$$

*Then, sampling $\tau$ uniformly at random from $[0,1]$ and letting $\bar{S}_N = \{i \in [n] : x_N^{\text{av}}(i) > \tau\}$ and $\bar{y}_N = y_N^{\text{av}}$, we have*

$$E_{\mathcal{U}_N}[D_\tau(\bar{S}_N, \bar{y}_N)] \leq \epsilon \tag{19}$$

*where*

$$D_\tau(S_k, y_k) = \max_{y \in \mathcal{Y}} E_\tau[f(S_k, y)] - \min_{S \subset [n]} f(S, y_k),$$

*and $\tau$ is the threshold used to round $x_N^{\text{av}}$, cf. (16).*

Proof of Theorem 3.2 is given in Appendix C. Theorem 3.2 shows that, in expectation, the average of the sequence generated by Algorithm 1 is an $\epsilon$-saddle point of $f^L$, and that the rounded pair $(\bar{S}_N, \bar{y}_N)$, which is computable from the averaged iterate by a single uniform sample of $\tau$, is an $\epsilon$-saddle point of $f$ in the sense of (19).

*Remark* 3.3 (Per-iteration cost of Algorithm 1). Computing subgradients twice (Definition 2.9) requires $2n$ evaluations of $f$, and evaluating Lovász extensions 4 times for the ZO oracle (see (14)) requires $4n$ evaluations. The total per-iteration cost is therefore $6n$ function evaluations. With $N = O(m^2 \epsilon^{-2})$ iterations (Theorem 3.2), the total sample complexity is $O(nm^2 \epsilon^{-2})$. The ZO estimator acts only on $y \in \mathbb{R}^m$, so the iteration count depends on $m$, not $n$. The factor $n$ is purely per-iteration cost from the Lovász extension and subgradient computations.

### 3.3. Online Settings

Now, we analyse the online min-max problem with submodular concave functions using a ZO framework. In the online setting, over a sequence of iterations $k = 0, 1, \ldots, N$ for some $N \in \mathbb{N}$, an online decision maker repeatedly selects a subset $S_k \subset [n]$ and $y_k \in \mathcal{Y}$. After choosing $(S_k, y_k)$, the cost is determined by a submodular-concave function $f_k : 2^{[n]} \times \mathbb{R}^m \to \mathbb{R}$, and the decision maker incurs the cost $f_k(S_k, y_k)$. To proceed, following (Zhang et al., 2022) we need to define a notion of optimality in this problem setting.

**Definition 3.4.** The online duality gap of the decision maker is defined as:

$$\text{Dual-Gap}_N := \sum_{k=0}^{N} \max_{y \in \mathcal{Y}} f_k(S_k, y) - \min_{S \subset [n]} f_k(S, y_k).$$

If the sets $S_k$ or $y_k$ are chosen by a randomised algorithm, the expected regret over the randomness in the algorithm is considered. Similar to the offline setting, we can define the online restricted gap as follows.

**Definition 3.5.** The online restricted gap of the decision maker is defined as:

$$\text{RD-Gap}_N := \sum_{k=0}^{N} f_k(S_k, y_k^*) - f_k(S_k^*, y_k),$$

where $(S_k^*, y_k^*)$ is a saddle point of $f_k$.

Definition 3.5 is also known as the online duality gap and used in works such as (Meng et al., 2025). For the analysis of the online setting, following (Zinkevich, 2003), we need to define the total path length of a sequence $(z_0, \ldots, z_N)$ as

$$P_N(z_0, \ldots, z_N) = \sum_{i=1}^{N} \|z_i - z_{i-1}\|, \tag{20}$$

which allows us to present our main result for this section. Also, We denote the Lovász extension of $f_k$ with respect to the first variable by $f_k^L$. Similar to the offline case, $f_k^L$ is Lipschitz continuous with $L_{0x,k}$. Moreover, based on $f_k^L$ being Lipschitz continuous, we have $|f_k^L(x,y)| \leq M_k$ for all $x \in [0,1]^n$ and $y \in \mathcal{Y}$, where $M_k$ is a positive scalar.

Next, we need to alter the random oracle defined in (14) to adapt it to this setting. Here we assume that before each function alteration from $f_k$ to $f_{k+1}$, we have enough time for at least four function queries (a discussion on relaxing this assumption to three function queries is provided in Appendix D). As mentioned after (11) and before (14), we know that $\nabla f_{\mu,k}^L(y) = E_u[\frac{f_k^L(y+\mu u)}{\mu}u]$ and $E[u] = 0$. Thus, we can define

$$g_{\mu_y,k}(x,y) = \frac{1}{\mu}(f_k^L(x, y + \mu u) - f_k^L(x,y))u, \quad (21)$$

with $E_u[g_{\mu,k}(x,y)] = \nabla_y f_{\mu,k}^L(x,y)$. In the online case, we consider Algorithm 1 replacing $G(z_k)$ and $G(\hat{z}_k)$ with

$$G_k(z_k) = \begin{bmatrix} g_{x,k}(z_k) \\ -g_{\mu_y,k}(z_k) \end{bmatrix}, \ G_k(\hat{z}_k) = \begin{bmatrix} g_{x,k}(\hat{z}_k) \\ -g_{\mu_y,k}(\hat{z}_k) \end{bmatrix} \quad (22)$$

and similar to (15), we define

$$\hat{F}_k(z) = \begin{bmatrix} g_{x,k}(z) \\ -\nabla_y f_{k_\mu}^L(z) \end{bmatrix}, \ F_k(z) = \begin{bmatrix} g_{x,k}(z) \\ -g_{y,k}(z) \end{bmatrix}, \quad (23)$$

where $\nabla_y f_{k_\mu}^L$ is the gradient of the Gaussian smoothed version of $f_k^L$ with respect to $y$, $g_{x,k}$ is the subgradient of $f_k^L$ with respect to $x$, and $g_{y,k}$ is the supergradient of $f_k^L$ with respect to $y$. Now, we can state the main theorem of this section showing that the sequence generated by Algorithm 1, in expectation, achieves an $O(\sqrt{N(1+\bar{P})})$ online duality gap, where $\bar{P}$ is a known upper bound on the path length $P_N$ defined below.

**Theorem 3.6.** *For $k \in [N]$, let $f_k : 2^{[n]} \times \mathbb{R}^m \to \mathbb{R}$ satisfy Assumptions 2.6 and 3.1 and let $f_k^L$ be their corresponding Lovász extension with respect to the first variable with $z_k^*$ as their saddle points. Let $N \geq 0$ be the number of iterations, $L_0 = \min\{\max_{k \in [N]}\{L_{0x,k}\}, 4\max_{k \in [N]}\{M_k\}\}$, $L_{0y} = \max_{k \in [N]}\{L_{0y,k}\}$, $t_k = t = 1$, $D_y$ be diameter of compact convex set $\mathcal{Y}$, $\mu > 0$ be the smoothing parameter in (11) and $\mathcal{U}_k = [u_0, \hat{u}_0, u_1, \hat{u}_1, \cdots, u_k, \hat{u}_k]$, $k \in [N]$. Moreover, let $\{z_k\}_{k \geq 0}$ and $\{\hat{z}_k\}_{k \geq 0}$ be the sequences generated by Algorithm 1. Then, for any iteration $N$, with constant step sizes $h_{1,k} = h_1$ and $h_{2,k} = h_2$, we have*

$$\frac{1}{N+1}E_{\mathcal{U}_N}[\text{RD-Gap}_N^L] \leq \frac{e_0^2}{2h_2(N+1)} + \mu L_{0y}m^{\frac{1}{2}}$$
$$+ (\frac{h_2}{2} + h_1)(L_0^2 + L_{0y}^2(m+4)^2) + \frac{3D_z P_N^*}{2h_2(N+1)}, \quad (24)$$

*where $\text{RD-Gap}_N^L = \sum_{k=0}^N f_k^L(\hat{x}_k, y_k^*) - f_k^L(x_k^*, \hat{y}_k)$ and $e_0 = \|z_0 - z_0^*\|, P_N^* = P_N(z_0^*, \ldots, z_N^*), D_z = \sqrt{n + D_y^2}$.*

*Additionally, the duality gap satisfies*

$$\frac{1}{N+1}E_{\mathcal{U}_N}[\text{Dual-Gap}_N^L] \leq \frac{E_{\mathcal{U}_N}[\bar{e}_0^2]}{2h_2(N+1)} + \mu L_{0y}m^{\frac{1}{2}}$$
$$+ (\frac{h_2}{2} + h_1)(L_0^2 + L_{0y}^2(m+4)^2) + \frac{3D_z E_{\mathcal{U}_N}[\bar{P}_N]}{2h_2(N+1)}, \quad (25)$$

*where $\bar{e}_0 = \|z_0 - \bar{z}_0\|, \bar{P}_N = P_N(\bar{z}_0, \ldots, \bar{z}_N)$,*

$$\bar{z}_k = \left(\arg\min_{x \in [0,1]^n} f_k^L(x, \hat{y}_k), \arg\max_{y \in \mathcal{Y}} f_k^L(\hat{x}_k, y)\right),$$

$$\text{Dual-Gap}_N^L = \sum_{k=0}^N \max_{y \in \mathcal{Y}} f_k^L(\hat{x}_k, y) - \min_{x \in [0,1]^n} f_k^L(x, \hat{y}_k).$$

*Moreover, let $\bar{P} \in \mathbb{R}_{\geq 0}$ be a known deterministic constant such that $\bar{P}_N \leq \bar{P}$ almost surely, let $\mu \leq \frac{1}{L_{0y}m^{\frac{1}{2}}(N+1)^{\frac{1}{2}}}$,*

$$h_1 \leq \frac{1}{(L_0^2 + L_{0y}^2(m+4)^2)^{\frac{1}{2}}(N+1)^{\frac{1}{2}}},$$

$$h_2 = \frac{(D_z^2 + 3D_z\bar{P})^{\frac{1}{2}}}{(L_0^2 + L_{0y}^2(m+4)^2)^{\frac{1}{2}}(N+1)^{\frac{1}{2}}}.$$

*Then, it holds that*

$$E_{\mathcal{U}_N}[\text{Dual-Gap}_{\tau,N}] \leq O(\sqrt{N(1+\bar{P})}) \quad (26)$$

*where $\tau$ is the threshold defined in (16) and*

$$\text{Dual-Gap}_{\tau,N} = \sum_{k=0}^N \max_{y \in \mathcal{Y}} E_\tau[f_k(\hat{S}_k, y)] - \min_{S \subset [n]} f_k(S, \hat{y}_k).$$

A proof of Theorem 3.6 can be found in Appendix C. The sequence $\bar{z}_k$, and hence $\bar{e}_0$ and $\bar{P}_N$, are random, as they depend on the iterates $\hat{z}_k$. This is why they appear inside the expectations in (25). The deterministic bound $\bar{P}$ used to set the step size $h_2$ always exists, since the problem is constrained, while sharper problem-dependent bounds may be available. Theorem 3.6 shows that in the expectation sense, the average of the (online) duality gap of $f^L$ and the average of the dual gap Dual-Gap$_{\tau,N}$ in the sequence generated by Algorithm 1 converge to zero asymptotically, provided that the path-length bound $\bar{P}$ grows sublinearly in $N$. In the next section, we will verify the theoretical findings through numerical examples.

## 4. Numerical Example

In this section, we will illustrate the theoretical results given in Section 3 through numerical examples. More examples and explanations are given in Appendix E[1].

---

[1]All the relevant codes are publicly available at https://github.com/amirali78frz/Minimax_projects.git.

*Table 1.* Comparison between the ZO online adversarial segmentation method and supervised and semi-supervised trained U-Net

| Property | Algorithm 1 | U-Net (supervised) | U-Net (semi-supervised) |
|---|---|---|---|
| Model type | Custom ZO optimiser | U-Net (CNN) | U-Net (CNN) |
| Number of parameters | $\sim 2.5$k (Optimisation variables) | $\sim$1.06M (Net weights) | $\sim$1.06M (Net weights) |
| Training requirement | **No pre-training** | Supervised on GT | Semi-supervised on seeds |
| Supervision | **Semi-supervised** | Supervised | **Semi-supervised** |
| Input | Seeds + image | **Image only** | Seeds + image |
| Adversarial robustness | **Yes (robust by design)** | No (standard CNN) | No (standard CNN) |
| Real-time capability | $\sim$80 fps | $\sim$80 fps (CPU) | $\sim$80 fps (CPU) |
| Ground truth needed | **Seeds** | Exact GT | **Seeds** |
| Average IoU | **0.975** | 0.905 | 0.847 |
| Average precision | **0.986** | 0.910 | 0.854 |
| Average recall | 0.989 | **0.994** | 0.991 |
| Average f1 score | **0.987** | 0.950 | 0.917 |
| Memory footprint | $\sim$ **0.05MB (state variables)** | $\sim$ 8MB | $\sim$ 8MB |
| Theoretical Guarantees | **Yes** | No | No |

## 4.1. Offline Adversarial Image Segmentation

In this section, we analyse the adversarial attack on semi-supervised image segmentation. The semi-supervised image segmentation through minimising a submodular cost function has been studied before, and its details can be found in Appendix E.1. Here, we extend the problem to the adversarial semi-supervised image segmentation, which can be formulated and solved through a min-max problem. We introduce the adversary vector $y \in \mathbb{R}^{|\mathcal{S}|}$, where $\mathcal{S}$ denotes the set of seed pixels (we reserve $S$ for the set variable of (1)), $y_s \in [0, 1]$, and $\sum_{s \in \mathcal{S}} y_s \leq \rho$. Vector $y$ can be viewed as trust weights of the given seeds, and $y_s = 1$ means a fully trusted seed for correction, and $y_s = 0$ means an ignored seed. Here, $\rho$ indicates the maximum number of seeds that can be trusted for correction. Thus, a lower $\rho$ means more uncertainty in the given seeds. The minimax problem cost function is defined in Appendix E.1. We solve this minimax problem using Algorithm 1. In our numerical experiments, we consider synthetic $50 \times 50$ noisy grayscale images consisting of two segments. A total of 20 foreground seeds and 20 background seeds are sampled uniformly at random from the corresponding regions. In Figure 1 the results of segmentation and convergence of the Lovász extension of the objective function for $\rho = 7$ is shown. The results demonstrate the recovered segmentation coincides with the ground truth. More explanation on the choice of $\rho$ and its effects is given in Appendix E.1.

## 4.2. Online Adversarial Image Segmentation

Here, we solve the online setting of the problem introduced in Section 4.1, where we have a video input where the shapes and clusters will move and rotate. Thus, edge weights, defined in Appendix E.1, change constantly, leading to an online problem. We solve the problem using Algorithm 1. In

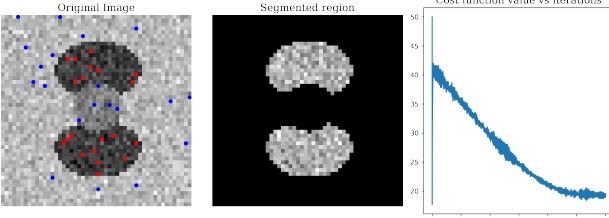

*Figure 1.* Offline Image Segmentation

this case, we consider a synthetic noisy 3-minute 60 frames per second (fps) video as the input, where each frame is a $50 \times 50$ image with 50 seeds for each cluster. The clusterings at times $t = 60s$ and $t = 120s$, are given in Figure 2.

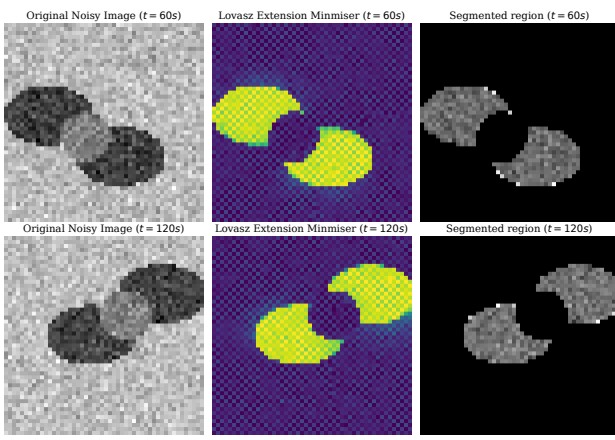

*Figure 2.* Online image segmentation under adversarial attack.

Algorithm 1 successfully performs online image segmentation despite continuously changing edge weights and adversarial perturbations. Importantly, the algorithm operates

in a fully online manner, performing a single extragradient update per incoming frame, and requires only the current state variables to be stored. We implemented a U-Net for supervised segmentation with 1,058,977 parameters, trained on 3000 samples of $50 \times 50$ images generated by random rotations and shifts of a base image, using ground-truth (GT) cluster labels. We also trained a semi-supervised U-Net with 1,059,841 parameters and a 3-channel input (image plus cluster seeds) to match the input of Algorithm 1, but without an adversarial. Both U-nets are trained and evaluated without adversaries. All methods were evaluated on the same input video and performed clustering in real time. The average IoU (intersection over union or Jaccard index) (Rota Bulo et al., 2017), precision, recall, and F1-score (Manning et al., 2008) over all input frames are reported in Table 1. As shown, even with adversaries, Algorithm 1 outperforms both supervised and semi-supervised U-Nets despite requiring no dataset or pre-training. Its independence from pre-training, real-time performance in unseen environments, and low memory footprint make it well-suited for embedded systems.

## 5. Conclusions and Future Work

In this study, we considered min-max and max-min problems with possibly non-differentiable, submodular-concave cost functions in both offline and online settings. We considered an algorithm exploiting the extragradient framework. The algorithm uses the subgradient of the Lovász extension of the objective function with respect to the minimiser and uses a Gaussian smoothing random oracle to estimate the smoothed function gradient with respect to the maximiser to update the estimates. We investigated the framework's performance in solving this problem. First, we considered the offline setting and showed that the algorithm, in the expectation sense, converges to an $\epsilon$-saddle point solution in $O(nm^2\epsilon^{-2})$ function calls. Then, we considered the online setting. We showed that, in the expectation sense, it achieves an online duality gap of $O(\sqrt{N(1 + \bar{P})})$, where $\bar{P}$ upper-bounds the path length of the sequence of optimal decisions. Numerical examples demonstrating the theoretical results were presented in the end. One possible future research topic is to explore the relations between solving (7) and finding an equilibria that might have a mixed strategy Nash equilibrium interpretation for a game related to the minimax problem with a submodular-concave cost function. A further possible direction is to investigate the use of a ZO algorithm in solving the minimax problem with submodular-submodular cost functions and to extend the results to weakly submodular functions. Another interesting direction is to extend the results for unbounded constraint sets.

## Acknowledgements

This work was supported by the Australian Research Council through a Discovery Project under Grant DP250101763 and the United States Air Force Office of Scientific Research under Grants FA9550-23-1-0424 and FA2386-24-1-4014.

## Impact Statement

This paper presents work whose goal is to advance the field of Machine Learning. There are many potential societal consequences of our work, none which we feel must be specifically highlighted here.

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

## A. Complementary Materials on Submodular Functions

In this section, we present complementary definitions and lemmas related to submodular functions and their Lovász extensions. The following lemma summarises a set of well-known properties of the Lovász extension function shown in (Lovász, 1983; Fujishige, 2005; Bach, 2013; Ando, 2002).

**Lemma A.1.** *Let $\bar{f} : 2^{[n]} \to \mathbb{R}$ be a submodular function and $\bar{f}^L$ be its Lovász extension. Then, we have that (i) $\bar{f}^L$ is convex, (ii) $\bar{f}^L$ is piecewise linear, (iii) $\bar{f}^L$ is positively homogeneous (i.e., for each $c > 0$, we have $\bar{f}^L(cx) = c\bar{f}^L(x)$), (iv) $\min_{S \subset [n]} \bar{f}(x) = \min_{x \in \{0,1\}^n} \bar{f}^L(x) = \min_{x \in [0,1]^n} \bar{f}^L(x)$, and (v) the set of minimisers of $\bar{f}^L(x)$ in $[0,1]^n$ is the convex hull of the set of minimisers of $\bar{f}^L(x)$ in $\{0,1\}^n$.*

Moreover, the subgradient of the Lovász extension of a submodular function satisfies the following property.

**Lemma A.2** ((Hazan & Kale, 2012), Lem 8)**.** *For all $x \in [0,1]^n$, the subgradients $g$ of the Lovász extension $f^L : [0,1]^n \to [-M, M]$, $M \in \mathbb{R}_{\geq 0}$, of a submodular function are bounded by*

$$\|g\| \leq 4M.$$

## B. Examples of Submodluar-Concave Functions with and without Saddle Points

In the following, we present two simple examples for the case when $\max_{y \in \mathcal{Y}} \min_{S \subset [n]} f(S, y) < \min_{S \subset [n]} \max_{y \in \mathcal{Y}} f(S, y)$, i.e., $f(S, y)$ has no saddle point and for the case that $\max_{y \in \mathcal{Y}} \min_{S \subset [n]} f(S, y) = \min_{S \subset [n]} \max_{y \in \mathcal{Y}} f(S, y)$, i.e., $f(S, y)$ has a saddle point.

*Example* B.1. Let $f : [1] \times [0,1] \to \mathbb{R}$ be defined as

$$f(\emptyset, y) = 1 - y, \qquad f(\{1\}, y) = y.$$

Using the second inequality in Definition 2.3, one can verify that $f(S, y)$ is submodular with respect to $S$. Moreover, $f(S, y)$ is concave with respect to $y$. Through direct calculation, we observe that

$$\min_{S \subset [1]} \max_{y \in [0,1]} f(S, y) = 1$$

and

$$\max_{y \in [0,1]} \min_{S \subset [1]} f(S, y) = \max_{y \in [0,1]} \min\{y, 1 - y\} = \tfrac{1}{2}.$$

Thus, $\max_{y \in [0,1]} \min_{S \subset [1]} f(S, y) < \min_{S \subset [1]} \max_{y \in \in [0,1]} f(S, y)$ and $f(S, y)$ has no saddle point. Now, considering (Bach, 2013) Def 3.1 and calculating the Lovász extension of $f(S, y)$ with respect to $S$, we get

$$f^L(x, y) = 2xy - x - y + 1.$$

Moreover, we can see that

$$\min_{x \in [0,1]} \max_{y \in [0,1]} f^L(x, y) = \max_{y \in [0,1]} \min_{x \in [0,1]} f^L(x, y) = \tfrac{1}{2},$$

which is attained at $x^* = y^* = \tfrac{1}{2}$.

*Remark* B.2. Even though the function discussed in Example B.1 does not have a saddle point, the function admits a generalised saddle-point analogue to the so-called mixed strategy Nash equilibrium (Osborne & Rubinstein, 1994), Def. 32.3. To see this, let $p = \Pr(S = \{1\})$. Then

$$\begin{aligned}
E_p[f(S, y)] &= pf(\{1\}, y) + (1 - p)f(\emptyset, y) \\
&= py + (1 - p)(1 - y) \\
&= 2py - p - y + 1.
\end{aligned}$$

Thus,

$$\min_{p \in [0,1]} \max_{y \in [0,1]} E_p[f(S, y)] = \max_{y \in [0,1]} \min_{p \in [0,1]} E_p[f(S, y)] = \tfrac{1}{2},$$

which is attained at $p^* = y^* = \tfrac{1}{2}$. Studying the relation between solutions of (7) and finding mixed strategy Nash equilibria of games with submodular-concave cost functions will be the focus of future work.

*Example* B.3. To give an example of a submodular-concave function having a saddle point, consider $f : [1] \times [0, 1] \to \mathbb{R}$ defined through

$$f(\emptyset, y) = 0.4 - (y - 0.5)^2, \qquad f(\{1\}, y) = 1.$$

We can see that

$$\max_{y \in [0,1]} \min_{S \subset [1]} f(S, y) = \min_{S \subset [1]} \max_{y \in [0,1]} f(S, y) = 0.4, \tag{27}$$

which is attained at the saddle point $S^* = \emptyset$ and $y^* = 0.5$. Calculating the the Lovász extension of $f(S, y)$ with respect to $S$, we get

$$f^L(x, y) = 0.4 + 0.6x + (x - 1)(y - 0.5)^2$$

and

$$\min_{x \in [0,1]} \max_{y \in [0,1]} f^L(x, y) = \max_{y \in [0,1]} \min_{x \in [0,1]} f^L(x, y) = 0.4,$$

which is attained at $x^* = 0$ and $y^* = 0.5$. This shows that in this example, as the minimiser of $f^L$ is on a vertex, we can solve (7) instead of (1).

## C. Proof of Lemmas, Propositions, and Theorems

In this section, the proofs of lemmas, propositions, and theorems presented in the main sections of this work are given. First, the proof of Lemma 2.7 is given.

*Proof of Lemma 2.7.* Since $f(\cdot, y)$ is submodular for all $y \in \mathbb{R}^m$, we can define the Lovász extension $f^L(\cdot, y)$ through Definition 2.4. From Lemma A.1, we know that $f^L(\cdot, y)$ is convex for all $y \in \mathbb{R}^m$. For any fixed $x \in [0, 1]^n$, consider $f^L(x, \cdot)$. Then, from (5), we can see that $f^L(x, \cdot)$ is a convex combination of concave functions. Thus, $f^L$ is convex-concave.

To prove the second statement, for $x \in [0, 1]^n$, order the components in decreasing order $x_{j_1} \geq \cdots \geq x_{j_n}$, where $(j_1, \ldots, j_n)$ is a permutation. Then, from Definition 2.4 applied with the chain $\emptyset = A_0 \subset \{j_1\} \subset \{j_1, j_2\} \subset \cdots \subset [n]$ and the coefficients $\lambda_0 = 1 - x_{j_1}$, $\lambda_k = x_{j_k} - x_{j_{k+1}}$ for $k \in [n-1]$, and $\lambda_n = x_{j_n}$ (which are non-negative and sum to one), the Lovász extension of $f(\cdot, y)$, $y \in \mathbb{R}^m$, can be written as

$$f^L(x, y) = f(\emptyset, y)(1 - x_{j_1}) + \sum_{k=1}^{n-1} f(\{j_1, \ldots, j_k\}, y)(x_{j_k} - x_{j_{k+1}}) + f([n], y)x_{j_n}. \tag{28}$$

Note that (28) reduces to (Bach, 2013, Def. 3.1) when the set function is normalised, i.e., $f(\emptyset, y) = 0$; the first term accounts for the general case, since here $f(\emptyset, y)$ may be nonzero and may depend on $y$. Now, sampling a threshold $\tau \in [0, 1]$ uniformly at random and letting $S_\tau = \{i \in [n] : x(i) > \tau\}$, we have

$$\begin{aligned} E_\tau[f(S_\tau, y)] &= \int_0^1 f(S_\tau, y)\mathrm{d}\tau \\ &= f(\emptyset, y)(1 - x_{j_1}) + \sum_{k=1}^{n-1} f(\{j_1, \ldots, j_k\}, y)(x_{j_k} - x_{j_{k+1}}) + f([n], y)x_{j_n} \\ &= f^L(x, y). \end{aligned}$$

Here, the first equality follows from the standard definition of expectation with respect to a random variable sampled from a uniform distribution between $[0, 1]$ (Dekking et al., 2005), the second equality follows from direct calculation of the integral and the fact that for $\tau \in [x_{j_{k+1}}, x_{j_k})$, $S_\tau = \{j_1, \ldots, j_k\}$, for $\tau \in [x_{j_1}, 1]$, $S_\tau = \emptyset$, and for $\tau \in [0, x_{j_n})$, $S_\tau = [n]$, so that $f(S_\tau, y)$ is piecewise constant in $\tau$, and the third equality follows from (28). Thus for all $y \in \mathbb{R}^m$ we have $E_\tau[f(S_\tau, y)] = f^L(x, y)$ and $\max_{y \in \mathcal{Y}} f^L(x, y) = \max_{y \in \mathcal{Y}} E_\tau[f(S_\tau, y)]$.

$\square$

In the following, the proof of Proposition 2.10 is given.

*Proof of Proposition 2.10.* Let $f : 2^{[n]} \times \mathbb{R}^m \to \mathbb{R}$ be a submodular-concave function and $f^L$ be the Lovász extension of $f$ with respect to the first variable. Considering (7), the fact that $f^L(x, y)$ is convex-concave, the constraints define convex and compact sets, and Sion's theorem (Sion, 1958),

$$\min_{x \in [0,1]^n} \max_{y \in \mathcal{Y}} f^L(x, y) = \max_{y \in \mathcal{Y}} \min_{x \in [0,1]^n} f^L(x, y) \tag{29}$$

holds, which guarantees that a saddle point for $f^L$ exists. Moreover, considering Lemma A.1, we know $\min_{x \in [0,1]^n} f^L(x, y) = \min_{S \subset [n]} f(S, y)$. Thus, we have

$$\max_{y \in \mathcal{Y}} \min_{x \in [0,1]^n} f^L(x, y) = \max_{y \in \mathcal{Y}} \min_{S \subset [n]} f(S, y),$$
$$\min_{x \in [0,1]^n} \max_{y \in \mathcal{Y}} f^L(x, y) = \max_{y \in \mathcal{Y}} \min_{S \subset [n]} f(S, y). \tag{30}$$

Moreover, as $\{0, 1\}^n \subset [0, 1]^n$, we know that

$$\min_{x \in [0,1]^n} \max_{y \in \mathcal{Y}} f^L(x, y) \le \min_{x \in \{0,1\}^n} \max_{y \in \mathcal{Y}} f^L(x, y), \tag{31}$$

and

$$\min_{x \in \{0,1\}^n} \max_{y \in \mathcal{Y}} f^L(x, y) = \min_{S \subset [n]} \max_{y \in \mathcal{Y}} f(S, y), \tag{32}$$

as $f^L$ and $f$ coincide on vertices. Thus, in general, we have

$$\max_{y \in \mathcal{Y}} \min_{S \subset [n]} f(S, y) \le \min_{S \subset [n]} \max_{y \in \mathcal{Y}} f(S, y). \tag{33}$$

Thus, in general existence of a saddle point for $f$ is not guaranteed. We either need to assume the existence of a saddle point for $f$ or investigate conditions such that

$$\min_{x \in [0,1]^n} \max_{y \in \mathcal{Y}} f^L(x, y) = \min_{x \in \{0,1\}^n} \max_{y \in \mathcal{Y}} f^L(x, y) \tag{34}$$

is satisfied, as if (34) holds, then combining (30), (31), and (32), we get

$$\max_{y \in \mathcal{Y}} \min_{S \subset [n]} f(S, y) = \min_{S \subset [n]} \max_{y \in \mathcal{Y}} f(S, y),$$

which guarantees the existence of saddle points for $f(S, y)$ and $\min_{S \subset [n]} \max_{y \in \mathcal{Y}} f(S, y) = \min_{x \in [0,1]^n} \max_{y \in \mathcal{Y}} f^L(x, y)$. $\square$

Next, the proof of Proposition 2.11 is given.

*Proof of Proposition 2.11.* To prove i), considering Lemma A.1.v), if $\zeta$ is a Lovász extension of a submodular function, then (34) holds immediately. Moreover, if the set of minimiser of $\zeta(x)$ over $x \in [0, 1]^n$ contains a vertex $x \in \{0, 1\}^n$, it is trivial to see that (34) holds. To prove the last statement, if for any $y \in \mathcal{Y}$, the set of minimiser of $f(\cdot, y)$ share a common vertex $x \in \{0, 1\}^n$, then it is guaranteed that the set of minimiser of $\zeta(x)$ contains a vertex and, consequently, (34) holds. $\square$

Next, we present the proof of Theorem 3.2.

*Proof of Theorem 3.2.* Let $r_k = \|z_k - z^*\|$. Using the non-expansiveness property of the projection to convex sets, we have

$$r_{k+1}^2 = \|\operatorname{Proj}_{\mathcal{Z}}(z_k - h_{2_k} G(\hat{z}_k)) - z^*\|^2$$
$$\le \|z_k - z^* - h_2 G(\hat{z}_k)\|^2$$

$$=r_k^2 - 2h_2\langle G(\hat{z}_k), z_k - z^*\rangle + h_2^2\|G(\hat{z}_k)\|^2. \tag{35}$$

Taking the expectation of (35) with respect to $\hat{u}_k$, we have

$$E_{\hat{u}_k}[r_{k+1}^2] \le r_k^2 - 2h_2\langle \hat{F}(\hat{z}_k), z_k - z^*\rangle + h_2^2 E_{\hat{u}_k}[\|G(\hat{z}_k)\|^2] \tag{36}$$

Moreover, considering the (Nesterov & Spokoiny, 2017) Thm. 4.1, Lemma A.2, and the fact that $f^L$ is Lipschitz, we have $\|g_x(\hat{z}_k)\|^2 \le L_0^2$ and $E_{\hat{u}_k}[\|g_{\mu_y}(\hat{z}_k)\|^2] \le L_{0y}^2(m+4)^2$. Thus

$$\begin{aligned} E_{\hat{u}_k}[\|G(\hat{z}_k)\|^2] &= \|g_x(\hat{z}_k)\|^2 + E_{\hat{u}_k}[\|g_{\mu_y}(\hat{z}_k)\|^2] \\ &\le L_0^2 + L_{0y}^2(m+4)^2. \end{aligned} \tag{37}$$

Additionally, note that the identity

$$\langle \hat{F}(\hat{z}_k), z_k - z^*\rangle = \langle \hat{F}(\hat{z}_k), \hat{z}_k - z^*\rangle + \langle \hat{F}(\hat{z}_k), z_k - \hat{z}_k\rangle. \tag{38}$$

holds. Now, let $s_k = \frac{1}{h_1}(z_k - \mathrm{Proj}_{\mathcal{Z}}(z_k - h_1 G(z_k)))$, which allows us to write

$$z_k - \hat{z}_k = h_1 s_k.$$

Using again the non-expansiveness property of the projection to convex sets, we get $\|s_k\| \le \|G(z_k)\|$. Considering the above and substituting (37) and (38) in (36), yields

$$\begin{aligned} \langle \hat{F}(\hat{z}_k), \hat{z}_k - z^*\rangle &\le \frac{r_k^2 - E_{\hat{u}_k}[r_{k+1}^2]}{2h_2} + \frac{h_2}{2}(L_0^2 + L_{0y}^2(m+4)^2) - \langle \hat{F}(\hat{z}_k), z_k - \hat{z}_k\rangle \\ &\le \frac{r_k^2 - E_{\hat{u}_k}[r_{k+1}^2]}{2h_2} + \frac{h_2}{2}(L_0^2 + L_{0y}^2(m+4)^2) - h_1\langle \hat{F}(\hat{z}_k), s_k\rangle \\ &\le \frac{r_k^2 - E_{\hat{u}_k}[r_{k+1}^2]}{2h_2} + \frac{h_2}{2}(L_0^2 + L_{0y}^2(m+4)^2) + h_1\|\hat{F}(\hat{z}_k)\|\|G(z_k)\| \end{aligned} \tag{39}$$

From Jensen's inequality, we know that $E_{u_k}[\|G(z_k)\|]^2 \le E_{u_k}[\|G(z_k)\|^2]$ and hence $E_{u_k}[\|G(z_k)\|] \le (L_0^2 + L_{0y}^2(m+4)^2)^{1/2}$. Moreover, to obtain an upper-bound for $\|\hat{F}(\hat{z}_k)\|$, Using the Jensen inequality and the fact that $\|\cdot\|^2$ is a convex function, we get

$$\|\hat{F}(\hat{z}_k)\|^2 \le \|E_{\hat{u}_k}[G(\hat{z}_k)]\|^2 \le E_{\hat{u}_k}[\|G(\hat{z}_k)\|^2] \le (L_0^2 + L_{0y}^2(m+4)^2)$$

i.e., $\|\hat{F}(\hat{z}_k)\| \le (L_0^2 + L_{0y}^2(m+4)^2)^{1/2}$. Considering this estimate and taking the expectation of (39) with respect to $u_k$, we have

$$E_{u_k}[\langle \hat{F}(\hat{z}_k), \hat{z}_k - z^*\rangle] \le \frac{E_{u_k}[r_k^2] - E_{\hat{u}_k}[r_{k+1}^2]}{2h_2} + (\tfrac{h_2}{2} + h_1)(L_0^2 + L_{0y}^2(m+4)^2). \tag{40}$$

In addition, from the convexity of $f^L(\cdot, y)$, we have

$$f^L(x, y) - f^L(x^*, y) \le \langle g_x(z), x - x^*\rangle, \tag{41}$$

and from the concavity of $f^L(x, \cdot)$ and considering (Nesterov & Spokoiny, 2017) Thm 2, we have

$$f^L(x, y^*) - f^L(x, y) \le \langle -\nabla_y f_\mu^L(z), y - y^*\rangle + \mu L_{0y} m^{1/2}. \tag{42}$$

Adding (41) and (42) together and letting $R^L(z) = f^L(x, y^*) - f^L(x^*, y)$, we have

$$R^L(z) \le \langle \hat{F}(z), z - z^*\rangle + \mu L_{0y} m^{1/2} \tag{43}$$

Substituting (43) in (40), we get

$$E_{u_k}[R^L(\hat{z}_k)] \le \frac{E_{u_k}[r_k^2] - E_{\hat{u}_k}[r_{k+1}^2]}{2h_2} + \mu L_{0y} m^{1/2} + (\tfrac{h_2}{2} + h_1)(L_0^2 + L_{0y}^2(m+4)^2). \tag{44}$$

Summing from $k = 0$ to $k = N$, taking the expectation with respect to $\mathcal{U}_{k-1}$, and dividing by $N + 1$, we arrive at (17).

To derive (18), consider the steps from (35) to (43) with an arbitrary $z \in \mathcal{Z}$ instead of the saddle point $z^*$. Then, we can derive the estimate

$$f^L(\hat{x}_k, y) - f^L(x, \hat{y}_k) \leq \langle \hat{F}(\hat{z}_k), \hat{z}_k - z \rangle + \mu L_{0y} m^{1/2}, \qquad \forall z = (x, y) \in \mathcal{Z}, \tag{45}$$

which holds pointwise for every realisation of the randomness. Next, we bound $\sup_{z \in \mathcal{Z}} \sum_{k=0}^N \langle \hat{F}(\hat{z}_k), \hat{z}_k - z \rangle$ in expectation. Fix $z \in \mathcal{Z}$. Using the non-expansiveness of the projection, for every realisation,

$$\langle G(\hat{z}_k), z_k - z \rangle \leq \frac{\|z_k - z\|^2 - \|z_{k+1} - z\|^2}{2h_2} + \frac{h_2}{2} \|G(\hat{z}_k)\|^2, \tag{46}$$

and, since $z_k - \hat{z}_k = h_1 s_k$ with $\|s_k\| \leq \|G(z_k)\|$,

$$\langle G(\hat{z}_k), \hat{z}_k - z \rangle \leq \frac{\|z_k - z\|^2 - \|z_{k+1} - z\|^2}{2h_2} + \frac{h_2}{2} \|G(\hat{z}_k)\|^2 + h_1 \|G(\hat{z}_k)\| \|G(z_k)\|. \tag{47}$$

Define the oracle noise $\Delta_k := G(\hat{z}_k) - \hat{F}(\hat{z}_k)$, which satisfies $E_{\hat{u}_k}[\Delta_k] = 0$ and, conditionally on $\hat{z}_k$,

$$E_{\hat{u}_k}[\|\Delta_k\|^2] = E_{\hat{u}_k}[\|G(\hat{z}_k)\|^2] - \|\hat{F}(\hat{z}_k)\|^2 \leq L_0^2 + L_{0y}^2(m + 4)^2.$$

Following the auxiliary-sequence technique of Nemirovski et al. (2009, Lem. 6.1, used there to prove Lem. 3.1), define $v_0 = z_0$ and $v_{k+1} = \text{Proj}_{\mathcal{Z}}(v_k + h_2 \Delta_k)$. The same projection argument as in (46) gives, for all $z \in \mathcal{Z}$,

$$\langle -\Delta_k, v_k - z \rangle \leq \frac{\|v_k - z\|^2 - \|v_{k+1} - z\|^2}{2h_2} + \frac{h_2}{2} \|\Delta_k\|^2. \tag{48}$$

Writing $\langle \hat{F}(\hat{z}_k), \hat{z}_k - z \rangle = \langle G(\hat{z}_k), \hat{z}_k - z \rangle - \langle \Delta_k, \hat{z}_k - v_k \rangle + \langle -\Delta_k, v_k - z \rangle$, summing (47) and (48) from $k = 0$ to $k = N$, and taking the supremum over $z \in \mathcal{Z}$ *after* the summation (both telescoping sums are bounded by $\bar{r}_0^2$ uniformly in $z$), we obtain

$$\sup_{z \in \mathcal{Z}} \sum_{k=0}^N \langle \hat{F}(\hat{z}_k), \hat{z}_k - z \rangle \leq \frac{\bar{r}_0^2}{h_2} + \sum_{k=0}^N \left[ \frac{h_2}{2} \|G(\hat{z}_k)\|^2 + h_1 \|G(\hat{z}_k)\| \|G(z_k)\| + \frac{h_2}{2} \|\Delta_k\|^2 \right] - \sum_{k=0}^N \langle \Delta_k, \hat{z}_k - v_k \rangle. \tag{49}$$

Both $\hat{z}_k$ and $v_k$ are measurable with respect to $\sigma(\mathcal{U}_{k-1}, u_k)$, while $E[\Delta_k \,|\, \mathcal{U}_{k-1}, u_k] = 0$; hence $E[\langle \Delta_k, \hat{z}_k - v_k \rangle] = 0$. Moreover, by the Cauchy–Schwarz inequality (applied twice, to the inner product and to the expectation of the product), $E[\|G(\hat{z}_k)\| \|G(z_k)\|] \leq (E[\|G(\hat{z}_k)\|^2])^{1/2}(E[\|G(z_k)\|^2])^{1/2} \leq L_0^2 + L_{0y}^2(m + 4)^2$. Taking the expectation of (49) therefore yields

$$E_{\mathcal{U}_N}\left[ \sup_{z \in \mathcal{Z}} \sum_{k=0}^N \langle \hat{F}(\hat{z}_k), \hat{z}_k - z \rangle \right] \leq \frac{\bar{r}_0^2}{h_2} + (N + 1)(h_2 + h_1)(L_0^2 + L_{0y}^2(m + 4)^2). \tag{50}$$

On the other hand, summing (45) from $k = 0$ to $k = N$ and using the convexity of $f^L(\cdot, y)$, the concavity of $f^L(x, \cdot)$, and Jensen's inequality, for every $z = (x, y) \in \mathcal{Z}$ it holds that

$$(N + 1)\left[ f^L(x_N^{\text{av}}, y) - f^L(x, y_N^{\text{av}}) \right] \leq \sum_{k=0}^N \left[ f^L(\hat{x}_k, y) - f^L(x, \hat{y}_k) \right] \leq \sum_{k=0}^N \langle \hat{F}(\hat{z}_k), \hat{z}_k - z \rangle + (N + 1)\mu L_{0y} m^{1/2}. \tag{51}$$

Taking the supremum over $z \in \mathcal{Z}$ on both sides of (51) (the supremum of the left-hand side is $(N+1)D^L(z_N^{\text{av}})$), dividing by $N+1$, taking the expectation, and substituting (50), we arrive at (18). Considering the chosen values for the hyperparameters and the upper-bound given in (18), we get

$$E_{\mathcal{U}_N}[D^L(z_N^{\text{av}})] \leq \epsilon.$$

Considering Lemma A.1, we know $\min_{S \subset [n]} f(S, y) = \min_{x \in [0,1]^n} f^L(x, y)$. From the projection step in Algorithm 1, we know that $\hat{x}_k \in [0, 1]^n$ for all $k$, and hence $x_N^{\text{av}} \in [0, 1]^n$ by convexity of the hypercube. Thus, considering Lemma 2.7, we have $f^L(x_N^{\text{av}}, y) = E_\tau[f(\bar{S}_N, y)]$ with $\bar{S}_N = \{i \in [n] : x_N^{\text{av}}(i) > \tau\}$. Moreover, $\tau$ and $\mathcal{U}_N$ are independent random variables. Thus, with $D_\tau(S_k, y_k) = \max_{y \in \mathcal{Y}} E_\tau[f(S_k, y)] - \min_{S \subset [n]} f(S, y_k)$, it follows that $D_\tau(\bar{S}_N, \bar{y}_N) = D^L(z_N^{\text{av}})$, and (19) follows.

$\square$

At last, the proof of Theorem 3.6 is given below.

*Proof of Theorem 3.6.* Let $e_k = \|z_k - z_k^*\|$. Then, we have

$$e_{k+1} = \|z_{k+1} - z_{k+1}^* + z_k^* - z_k^*\|$$
$$\leq \|z_{k+1} - z_k^*\| + \|z_{k+1}^* - z_k^*\|$$

and with $\|z_{k+1} - z_k^*\| \leq D_z$, where $D_z = \sqrt{n + D_y^2}$, it holds that

$$e_{k+1}^2 \leq \|z_{k+1} - z_k^*\|^2 + \|z_{k+1}^* - z_k^*\|^2 + 2D_z\|z_{k+1}^* - z_k^*\|. \tag{52}$$

Using the non-expansiveness property of projections on convex sets, we have

$$\|z_{k+1} - z_k^*\|^2 \leq \|\text{Proj}_{\mathcal{Z}}(z_k - h_{2_k}G_k(\hat{z}_k)) - z_k^*\|^2$$
$$\leq \|z_k - z_k^* - h_2 G_k(\hat{z}_k)\|^2 \tag{53}$$
$$\leq e_k^2 - 2h_2\langle G_k(\hat{z}_k), z_k - z_k^*\rangle + h_2^2\|G_k(\hat{z}_k)\|^2.$$

Combining (52) and (53), we get

$$e_{k+1}^2 \leq e_k^2 - 2h_2\langle G_k(\hat{z}_k), z_k - z_k^*\rangle + h_2^2\|G_k(\hat{z}_k)\|^2 + 3D_z\|z_{k+1}^* - z_k^*\|. \tag{54}$$

Taking the expectation with respect to $\hat{u}_k$ and similar to the steps (37) to (40) in proof of Theorem 3.2, we get

$$E_{u_k}[\langle \hat{F}_k(\hat{z}_k), \hat{z}_k - z_k^*\rangle] \leq \frac{E_{u_k}[e_k^2] - E_{\hat{u}_k}[e_{k+1}^2]}{2h_2} + (\tfrac{h_2}{2} + h_1)(L_0^2 + L_{0y}^2(m+4)^2) + \frac{3D_z}{2h_2}\|z_{k+1}^* - z_k^*\|. \tag{55}$$

Similar to (43), we get

$$f_k^L(\hat{x}_k, y_k^*) - f_k^L(x_k^*, \hat{y}_k) \leq \langle \hat{F}_k(\hat{z}_k), \hat{z}_k - z_k^*\rangle + \mu L_{0y}m^{1/2} \tag{56}$$

Now, Combining (55) and (56), summing from $k = 0$ to $k = N$, taking the total expectation with respect to $\mathcal{U}_N$, and dividing by $N + 1$, and letting $P_N^* = P_N(z_0^*, \ldots, z_N^*)$ and RD-Gap$_N^L = \sum_{k=0}^N f_k^L(\hat{x}_k, y_k^*) - f_k^L(x_k^*, \hat{y}_k)$, we arrive at (24).

To derive (25), we repeat the steps from (52) to (56) with $\bar{z}_k = (\arg\min_{x\in[0,1]^n} f_k^L(x, \hat{y}_k), \arg\max_{y\in\mathcal{Y}} f_k^L(\hat{x}_k, y))$ instead of $z_k^*$ and defining $\bar{e}_k = \|z_k - \bar{z}_k\|$, we get

$$E_{u_k}[\langle \hat{F}_k(\hat{z}_k), \hat{z}_k - \bar{z}_k\rangle] \leq \frac{E_{u_k}[\bar{e}_k^2] - E_{\hat{u}_k}[\bar{e}_{k+1}^2]}{2h_2} + (\tfrac{h_2}{2} + h_1)(L_0^2 + L_{0y}^2(m+4)^2) + \frac{3D_z}{2h_2}\|\bar{z}_{k+1} - \bar{z}_k\|. \tag{57}$$

and

$$f_k^L(\hat{x}_k, \bar{y}_k) - f_k^L(\bar{x}_k, \hat{y}_k) = \max_{y\in\mathcal{Y}} f_k^L(\hat{x}_k, y) - \min_{x\in[0,1]^n} f_k^L(x, \hat{y}_k)$$
$$\leq \langle \hat{F}_k(\hat{z}_k), \hat{z}_k - \bar{z}_k\rangle + \mu L_{0y}m^{1/2} \tag{58}$$

Now, combining (57) and (58), summing from $k = 0$ to $k = N$, taking the total expectation with respect to $\mathcal{U}_N$ (note that $\bar{z}_k$ is a deterministic function of $\hat{z}_k$, so the conditional-expectation steps remain valid, while $\bar{e}_0$ and $\bar{P}_N$ are random and appear in expectation), dividing by $N + 1$, letting $\bar{P}_N = P_N(\bar{z}_0, \ldots, \bar{z}_N)$ and Dual-Gap$_N^L = \sum_{k=0}^N \max_{y\in\mathcal{Y}} f_k^L(\hat{x}_k, y) - \min_{x\in[0,1]^n} f_k^L(x, \hat{y}_k)$, we arrive at (25). Considering the chosen values for the hyperparameters, together with $\bar{e}_0 \leq D_z$ and $E_{\mathcal{U}_N}[\bar{P}_N] \leq \bar{P}$, and the upper-bound given in (25), we get

$$E_{\mathcal{U}_N}[\text{Dual-Gap}_N^L] \leq (N+1)^{\frac{1}{2}}(1 + (L_0^2 + L_{0y}^2(m+4)^2)^{\frac{1}{2}}(1 + (D_z^2 + 3D_z\bar{P})^{\frac{1}{2}})) \tag{59}$$

or

$$E_{\mathcal{U}_N}[\text{Dual-Gap}_N^L] \leq O(\sqrt{N(1 + \bar{P})}).$$

Considering Lemma A.1, we know that $\min_{S\subset[n]} f_k(S, y) = \min_{x\in[0,1]^n} f_k^L(x, y)$. From the projection step in Algorithm 1, we get $\hat{x}_k \in [0,1]^n$ for all $k \in [N]$. Thus, considering Lemma 2.7, we have $f_k^L(\hat{x}_k, y) = E_\tau[f_k(\hat{S}_k, y)]$. Moreover, we know that $\tau$ and $u_k$ are independent random variables. Thus, considering above and letting Dual-Gap$_{\tau,N} = \sum_{k=0}^N \max_{y\in\mathcal{Y}} E_\tau[f_k(\hat{S}_k, y)] - \min_{S\subset[n]} f_k(S, \hat{y}_k)$, we get (26), which completes the proof.

$\square$

## D. Complementary Remarks on Results Given in Section 3

In this section, we provide some complementary explanation on the discussions and results given in Section 3. The next remark will explain the structure of Algorithm 1.

*Remark* D.1. For constructing a ZO algorithm, we could employ the random oracle in (12) with respect to $x$ and $y$ and not only with respect to $y$ as in Algorithm 1. However, in (Farzin et al., 2025c), the authors study the use of Gaussian random oracles for minimising a submodular function and show that, while the ZO method achieves similar performance to the subgradient method in minimising a submodular function and in expectation, the latter is at least twice as fast. This difference arises because computing a subgradient requires $n$ function evaluations, whereas computing the random oracle requires at least (with $t_k = 1$) $2n$ evaluations (calculating the Lovász extension requires $n$ function evaluations). This justifies why the random oracle is only applied to the $y$-component but not to the $x$-component in the algorithm developed in this paper.

The next remark gives more explanation about the choice of the oracle (12).

*Remark* D.2. If, instead of using (12), we employ either the central difference or the backward difference random oracle, defined respectively as $\hat{g}_\mu(y) = \frac{\tilde{f}(y+\mu u) - \tilde{f}(y-\mu u)}{2\mu} u$, or $\bar{g}_\mu(y) = \frac{\tilde{f}(y) - \tilde{f}(y-\mu u)}{\mu} u$, we can still derive the same bounds as those in Theorem 3.2. This follows from the fact that, for a Lipschitz continuous function, the expected squared norm of each of these oracles admits the same upper bound.

In section 3.3, we assume that we have enough time for at least four function evaluations between each function change from $f_k$ to $f_{k+1}$, for example, between each input frame in Section 4.2. Below, we explain how we can relax this assumption to have enough time for at least three function evaluations between each function change and how the results given in Section 3.3 will change.

Here, as considered in (Shames et al., 2019; Farzin et al., 2025c), we assume that the cost function can change between the function evaluations that are needed in each iteration of Algorithm 1. Thus we consider working with a family of submodular-concave functions $f_k : 2^{[n]} \times \mathbb{R}^m \to \mathbb{R}$ where $k \in \mathbb{N} \cup \{j + \frac{1}{2} | j \in \mathbb{N}\} = \{0, 1/2, 1, 3/2, \dots\}$. For simplicity, we denote $k + \frac{1}{2}$ by $k^+$ for all $k \in \mathbb{N}$. Next, we need to alter the random oracle defined in (14) to adapt it to this setting. Here we assume that before each function alteration from $f_k$ to $f_{k^+}$, we have enough time for at least three function queries. As mentioned after (11) and before (14), we know that $\nabla f^L_{\mu,k}(y) = E_u[\frac{f^L_k(y+\mu u)}{\mu}u]$ and $E[u] = 0$. Thus, we can define

$$g_{\mu_y,k}(x,y) = \tfrac{1}{\mu}(f^L_k(x, y+\mu u) - f^L_k(x,y))u, \tag{60}$$

$$g_{\mu_y,k^+}(x,y) = \tfrac{1}{\mu}(f^L_k(x, y+\mu u) - f^L_{k^+}(x,y))u, \tag{61}$$

with $E_u[g_{\mu,k^+}(x,y)] = \nabla_y f^L_{\mu,k}(x,y)$. We consider Algorithm 1 replacing $G(z_k)$ and $G(\hat{z}_k)$ with

$$G_k(z_k) = \begin{bmatrix} g_{x,k}(z_k) \\ -g_{\mu_y,k}(z_k) \end{bmatrix}, \; G_{k^+}(\hat{z}_k) = \begin{bmatrix} g_{x,k}(\hat{z}_k) \\ -g_{\mu_y,k^+}(\hat{z}_k) \end{bmatrix}. \tag{62}$$

In this case, we need an extra assumption. In particular, we need to assume that

$$|f^L_k(z) - f^L_{k^+}(z)| \le V, \qquad \forall z \in \mathbb{R}^d \tag{63}$$

where $V \in \mathbb{R}_{\ge 0}$ denotes a constant. This is a standard assumption in the literature of online ZO optimisation (Shames et al., 2019), and is a crucial assumption for controlling the variance of the random oracle.

The next remark explains more on how the results given in Theorem 3.6 will change with the assumption of having enough time for at least 3 function queries before each cost function alteration instead of 4 function calls.

*Remark* D.3. We know $E_u[G_{k^+}(z)] = \hat{F}(z)$ as $E_u[g_{\mu,k^+}(x,y)] = \nabla_y f^L_{\mu,k}(x,y)$ and thus, in expectation we can use (61) instead of (21) to get the same results as given in (26), but with one difference, we need (63) to be satisfied to be able to control the variance of $G_{k^+}(z)$. As we use two different functions $f_k$ and $f_{k^+}$ to calculate $g_{\mu,k^+}$, we can no longer use (Nesterov & Spokoiny, 2017) Thm 4.1 to bound $E_u[\|G_{k^+}(z)\|^2]$ and conclude that $E_{\hat{u}_k}[\|G_{k^+}(\hat{z}_k)\|^2] \le L_0^2 + L_{0y}^2(m+4)^2$. In this case, we can reformulate $g_{\mu_y,k^+}(x,y)$ as

$$g_{\mu_y,k^+}(x,y) = \tfrac{1}{\mu}(f^L_k(x, y+\mu u) - f^L_k(x,y) + f^L_k(x,y) - f^L_{k^+}(x,y))u. \tag{64}$$

Then similar to (Nesterov & Spokoiny, 2017) Thm 4.1, we can show that $E_u[\|g_{\mu_y,k+}(x,y)\|^2] \leq 2L_{0y}^2(m+4)^2 + \frac{2mV^2}{\mu^2}$ and thus

$$E_{\hat{u}_k}[\|G_{k+}(\hat{z}_k)\|^2] \leq L_0^2 + 2L_{0y}^2(m+4)^2 + \frac{2mV^2}{\mu^2} \tag{65}$$

holds. Hence, we can consider (65), follow the exact same steps of the proof of Theorem 3.6, and with proper choice of hyperparameters, conclude the same results as in (26).

The next remark explains more on the random oracle (61) and on the assumption of having enough time for at least 3 function queries before each cost function alteration.

*Remark* D.4. Similar to (Farzin et al., 2025c) Rem. 4, if in the update step for the online setting

$$\tilde{g}_{\mu_y,k+}(x,y) = \tfrac{1}{\mu}(f_{k+}^L(x,y+\mu u) - f_k^L(x,y))u$$

is used instead of (61), then we get $\nabla_y f_{\mu,k+}^L(x,y) = E_u[\tilde{g}_{\mu,k+}(x,y)]$. In this case, $\nabla_y f_{\mu,k+}^L(x)$ is not an unbiased estimator of $\nabla_y f_{\mu,k}^L(x)$ and we cannot use (Nesterov & Spokoiny, 2017) Thm. 2 directly and an extra assumption is necessary. In particular, we need to (63) to be satisfied. This means that the change of the functions' values during the sampling period in each iteration is upper-bounded. With this assumption, we can get the estimate

$$E_{\mathcal{U}_N}[\text{Dual-Gap}_{\tau,N}] \leq O(\sqrt{N(1+\bar{P})}) + O(NV), \tag{66}$$

which means that there exists a non-vanishing extra error that depends on the change in the functions' values over the sampling period. The same conclusions can be made about the results if we remove the assumption of having enough time for at least 3 function queries before the cost function alteration.

## E. Extra Numerical Examples and Explanations

In this section, we will give more explanations on the examples given in Section 4 and present two more examples to illustrate the theoretical results given in Section 3. All the tests have been run on a Dell Latitude 7430 Laptop with a 12th Gen Intel Core i7 CPU. [23].

### E.1. Offline Adversarial Image Segmentation

In this section, we analyse the adversarial attack on semi-supervised image segmentation. Previously, in works such as (Bach, 2013), the problem of offline semi-supervised image segmentation through minimisation of a submodular function has been explored and solved using different algorithms. The minimisation problem and its cost function are defined as the following. Given $p$ points in a certain set $V$ and weights $d : V \times V \to \mathbb{R}_{\geq 0}$ we intend to select a set $A \subset V$, which minimises the cut cost function

$$Cost(A) = d(A, V \setminus A)$$

where we denote $d(B,C) = \sum_{k \in B, \, j \in C} d(k,j)$ for any two sets $B, C$. Here we consider $50 \times 50$ images as set $V$, i.e., $p = 2500$. For this case, we convert the image into a graph, where each pixel is a node, edges connect 4 neighbours and edge weights measure similarity, i.e.,

$$d(k,j) = \exp\left(-\frac{(I_k - I_j)^2}{2\sigma_I^2} - \frac{(x_k - x_j)^2}{2\sigma_x^2}\right).$$

Here, $I_j$ and $x_j$ are the intensity and location of a pixel $j$, respectively, and $\sigma_x, \sigma_I \in \mathbb{R}_{>0}$ denote weighting parameters. For the following experiments, the parameters $\sigma_x = 1$ and $\sigma_I = 20$ are used. Above cost function is shown to be a submodular function in (Bach, 2013, Sec. 6). Pixels with similar intensities have large $d(k,j)$, discouraging cuts between them. The weight between two pixels is a decreasing function of their distance in the image and the difference of pixel intensities. As

---

[2]The Python code is publicly available at https://github.com/amirali78frz/Minimax_projects.git.

[3]You can find the Python package for zeroth-order solvers using random oracles at https://github.com/amirali78frz/ZoSolvers.git and https://pypi.org/project/ZoSolvers/

mentioned this problem is semi-supervised. Thus, we have seed pixels $s \in \mathcal{S}$, where $\mathcal{S}$ is the set of seeds(we reserve $S$ for the set variable of (1)) The segmentation is obtained by minimising the cost function regarding the labelling constraints.

Next, we extend the above problem to the adversarial semi-supervised image segmentation, which can be formulated and solved through a min-max problem. As mentioned above, the minimisation problem is constrained to the labels for the given seeds. In the adversarial setting, we introduce the adversary vector $y \in \mathbb{R}^{|\mathcal{S}|}$, where $y_s \in [0, 1]$ and $\sum_{s \in \mathcal{S}} y_s \leq \rho$. Then, vectors $y$ can be viewed as enforcement trust weights of the given seeds, $y_s = 1$ means a fully trusted seed for correction, $y_s = 0$ means an ignored seed and $\rho$ indicates the maximum number of seeds that can be trusted for correction. Thus, a lower $\rho$ means more uncertainty in the given seeds. Since the adversary maximises the objective, it selectively enforces the constraints of misclassified seeds, penalising the learner most. The learner selects the subset $A \subset V$ to minimise the overall cost, while the adversary maximises it. The resulting adversarial optimisation problem is

$$\min_{A \subset V} \max_{\substack{y \in [0,1]^{|\mathcal{S}|} \\ \sum_{s \in \mathcal{S}} y_s \leq \rho}} \left\{ d(A, V \setminus A) + \lambda \sum_{s \in \mathcal{S}} y_s \left| \mathbf{1}_A(s) - \ell_s \right| \right\},$$

where $\ell_s \in \{0, 1\}$ denotes the prescribed label of seed $s$, $\mathbf{1}_A(s)$ is the indicator of membership of $s$ in $A$, and $\lambda > 0$ is a penalty parameter weighting the influence of the seed constraints. We solve this minimax problem using Algorithm 1. In our numerical experiments, we consider synthetic $50 \times 50$ noisy grayscale images consisting of two segments. A total of 20 foreground seeds and 20 background seeds are sampled uniformly at random from the corresponding regions. We choose $\lambda = 5$, $h_1 = h_2 = 10^{-3}$, $\mu = 10^{-5}$, and $t_k = t = 20$.

In Figure 3, the results of segmentation and convergence of the Lovász extension of the objective function for three values of $\rho$ are shown. The results demonstrate a sharp transition with respect to the adversarial budget $\rho$. When $\rho$ exceeds a critical threshold (approximately $\rho \geq 5$ for this example), the recovered segmentation coincides with the ground truth. For $\rho$ below this threshold, no choice of algorithmic parameters yields a correct segmentation, as the adversary can selectively enforce an insufficient number of seeds, allowing the minimiser to disconnect the object. This behaviour highlights the intrinsic robustness–accuracy trade-off induced by the adversarial formulation and illustrates the role of $\rho$ as a measure of the minimum number of reliable seeds required to preserve global connectivity.

In Figure 3, the surge in the cost function during the transition period before convergence is an observed phenomenon in min-max optimisation. It arises because the extragradient method alternates between the minimiser pushing the cost down and the maximiser pushing it up, causing transient oscillations before the two players reach equilibrium. Importantly, our convergence guarantee, Theorem 3.2, is for the expected duality gap and the best generated candidate, not for monotonic decrease at every iteration. This is standard in the optimisation and ZO literature (Nesterov & Spokoiny, 2017; Farzin et al., 2025a). As can be seen in Figures 1 and 3, despite the transient surges, the algorithm ultimately produces correct segmentations that match the ground truth.

### E.2. Online Adversarial Image Segmentation

In this section, we solve the online setting of the problem introduced in Section 4.1, where we have a video input where the shapes and clusters move and rotate. Thus, edge weights, defined in Appendix E.1, change constantly, leading to an online problem. We solve the problem using Algorithm 1 with parameters $\lambda = 10$ $h_1 = h_2 = 3 \times 10^{-2}$, $\mu = 10^{-3}$, $t_k = 10$ and $\rho = 25$. In this case, we consider a synthetic noisy 3-minute 60 frames per second (fps) video as the input, where each frame is a $50 \times 50$ image with 50 seeds for each cluster. The seeds are resampled at every frame, uniformly from the corresponding noiseless regions. The segmentation using Algorithm 1 and a Python implementation can be done in real-time for videos up to 60 fps. The Lovász extension value versus iterations is given in Figure 4.

Figure 4 corresponds to the online setting, where the cost function $f_k$, changes at every iteration (each frame of the video introduces new edge weights due to the rotating and moving object), as well as the solution. A monotonically decreasing trend is not expected unless variations vanish, which is not the case here. The algorithm's goal is to track the changing optimum, not converge to a fixed point. Evidence of near-saddle-point solutions at each frame is provided by Table 1 and Figures 2 and 5, which show that the segmentation consistently matches the ground truth despite the changing cost function and adversarial perturbations. Moreover, the clusterings at times $t = 0$, $t = 60s$, $t = 120s$, and $t = 180s$ are shown in Figure 5. As can be seen, Algorithm 1 successfully performs online image segmentation despite continuously changing edge weights and adversarial perturbations. Importantly, the algorithm operates in a fully online manner, performing a single extragradient update per incoming frame, and requires only the current state variables to be stored. As a result, both memory

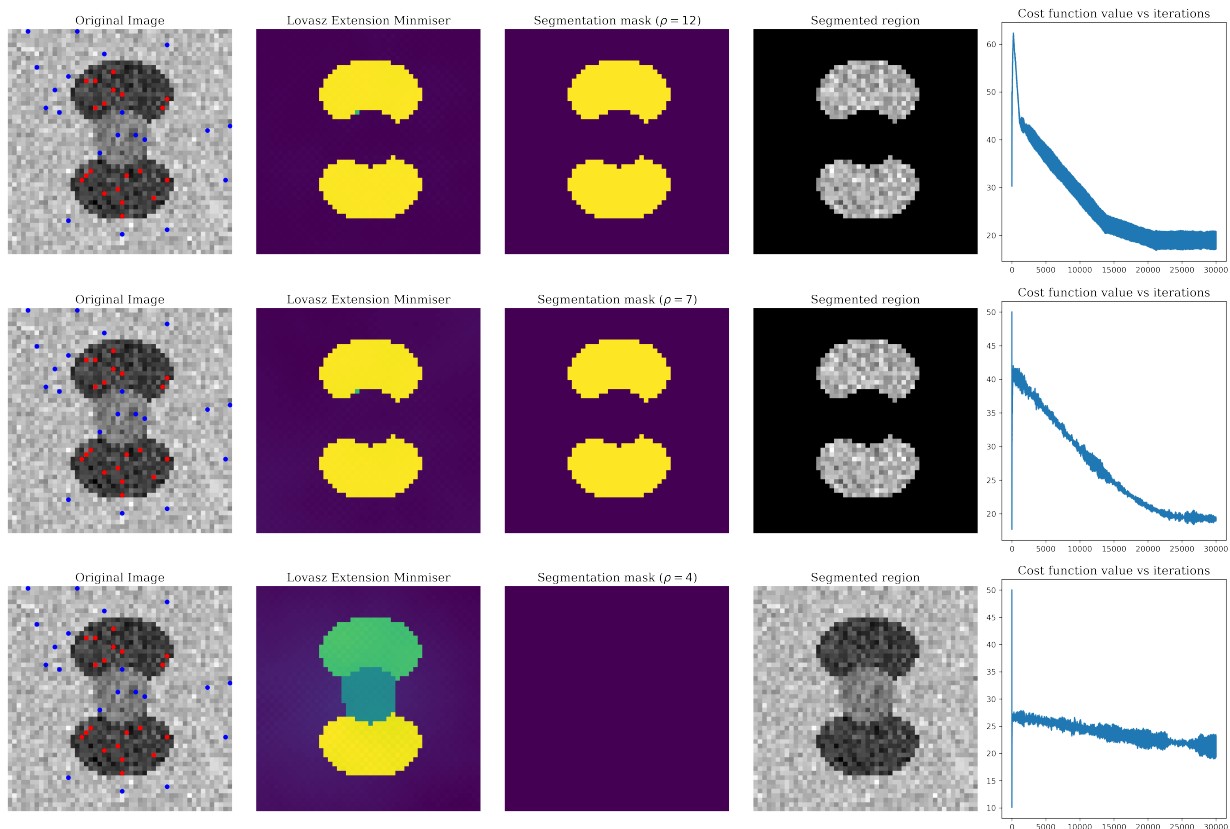

*Figure 3.* Offline Image Segmentation

usage and computational complexity scale linearly with the number of pixels and edges, yielding predictable per-frame latency. In our Python implementation, this allows real-time operation at 60 frames per second on standard CPU hardware.

This stands in contrast to deep neural network-based segmentation approaches, which typically require offline training on large datasets, substantial memory to store millions of learned parameters, and dedicated GPU acceleration to achieve comparable inference speeds. On embedded or resource-constrained robotic platforms without internet access, such requirements significantly limit deployability. While large deep models may achieve competitive accuracy under unconstrained settings, they fail to meet real-time and memory constraints in the online adversarial semi-supervised regime considered here. These results highlight the suitability of Algorithm 1 for autonomous systems requiring reliable, real-time segmentation without reliance on pretraining or external computational resources. Our comparison is not intended to replace deep learning in data-rich offline supervised settings, but to evaluate segmentation algorithms under strict online, adversarial, and resource-constrained conditions.

As a next example, we consider a U-Net network for supervised segmentation. The U-Net has $1058977$ parameters and is trained on a dataset of 3000 samples of $50 \times 50$ images obtained from random rotations and translations of a base image considered at $t = 0s$. The training motions are drawn from a family of rotations and translations whose ranges cover the motions encountered in the evaluation video, so the evaluation data lie within the training distribution. The training is supervised, and ground truth (GT) clustering for each data point is given to the network. Also, we have trained a U-Net with $1059841$ parameters and a 3-channel input, which includes the image and seeds for each cluster to have the same input as Algorithm 1. Thus, this network is trained semi-supervised (same number of seeds per frame as for Algorithm 1 but without adversary), but its setting is not adversarial. Both U-nets are trained and evaluated without adversaries. All the methods have been tested on the same input video and performed clustering in real time. The comparison of average IoU (intersection-over-union), average precision, average recall, and average F1 score across all input frames for all methods is reported in Table 1. For Algorithm 1, the per-frame discrete segmentation is obtained by thresholding the iterate at $0.5$ (the iterates are near-binary in this application) rather than by sampling the threshold $\tau$ uniformly at random, and its metrics are averaged over all frames after discarding the first two seconds (120 frames) of the video as an initial transient; the U-Net

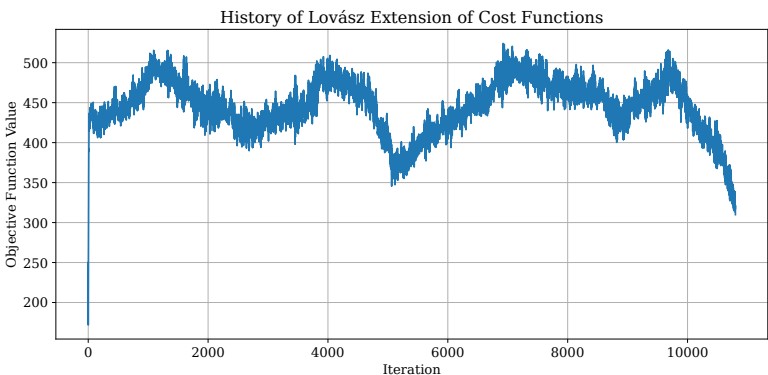

*Figure 4.* Lovász extension history over iterations for online image segmentation.

metrics are averaged over all frames. The results for Algorithm 1 are obtained by averaging over 10 independent runs. The variation across runs is on the order of $10^{-4}$ relative to the reported values, meaning a good consistency across the runs. The segmented region using the supervised and semi-supervised U-Net at different times is shown in Figure 6. It can be seen from Table 1 that without any need to datasets and pre-training, Algorithm 1 outperforms the trained supervised and semi-supervised U-Nets. It may be possible to improve the performance of both supervised and semi-supervised networks by tuning their architecture and hyperparameters, but still the performance of Algorithm 1 will be competitive. This independency of Algorithm 1 from pre-training and its ability to work in real-time in unseen environments while needing low memory, makes it suitable for embedded systems. We also note that the reported performance of Algorithm 1 in Table 1 depends on the choice of $\rho$. Larger values of $\rho$ (corresponding to more reliable seeds) lead to higher performance metrics, whereas smaller values of $\rho$ (indicating less trust in the given seeds) result in lower reported values. The parameter $\rho$ does not affect the U-Net baselines, as they are trained and evaluated without adversarial perturbations. One limitation of the proposed method is that the accuracy of segmentation and tuning of the parameters is sensitive to the rate of change in the cost functions over time, as it is expected and explained in Remarks D.4 and D.3.

### E.3. Offline Adversarial Attack on Semi-Supervised Clustering

In this section, we analyse the adversarial attack on semi-supervised clustering. Previously, in works such as (Bach, 2013; Farzin et al., 2025c), the problem of offline and online semi-supervised clustering through minimisation of a submodular function has been explored and solved using different algorithms. The minimisation problem and its cost function are defined as the following. Given $p \in \mathbb{N}$ data points in a certain set $V$, we intend to select a set $A \subset V$, which minimises the cost function

$$Cost(A) = I(f_A, f_{V \setminus A}) - \sum_{k \in A} \log \eta_k - \sum_{k \in V \setminus A} \log(1 - \eta_k),$$

where, $f_A$ and $f_{V \setminus A}$ are two Gaussian processes with zero mean and covariance matrices $K_{AA}$ and $K_{V \setminus A V \setminus A}$,

$$I(f_A, f_{V \setminus A}) = \tfrac{1}{2} \big( \mathrm{logdet}(K_{AA}) + \mathrm{logdet}(K_{V \setminus A V \setminus A}) - \mathrm{logdet}(K_{VV}) \big)$$

is the mutual information between the Gaussian processes, and $\eta_k$ is the probability of each element to be in set $A$. We consider each data point $x_i \in \mathbb{R}^2$ and sample each data point from the "two-moons" dataset (Bach, 2013). We consider 50 points and 10 randomly chosen labelled points. We impose $\eta_k \in \{0, 1\}$ for the labelled points and $\eta_k = \frac{1}{2}$ for the rest. We consider the Gaussian radial basis function kernel $k(x, y) = \exp(\frac{-\|x-y\|^2}{2\sigma^2})$ with $\sigma^2 = 0.05$ to obtain covariance matrices.

Here, we extend the above problem to the adversarial attack on semi-supervised clustering, which can be formulated and solved through a min-max problem. In this setting, an adversary perturbs the prior probability of the labelled data through an attack vector $y \in [0, 1]^{|M|}$, where $M$ denotes the set of labelled indices and $y_i = 1$ indicates that the label of the $i$-th labelled point is completely flipped. Vector $y$ can be viewed as the uncertainty of the labelled data. The learner, on the other hand, selects the subset $A \subset V$ to minimise the overall cost, while the adversary maximises it. The resulting adversarial optimisation problem is

$$\min_{A \subset V} \max_{y \in \mathcal{Y}_\rho} f(A, y), \tag{67}$$

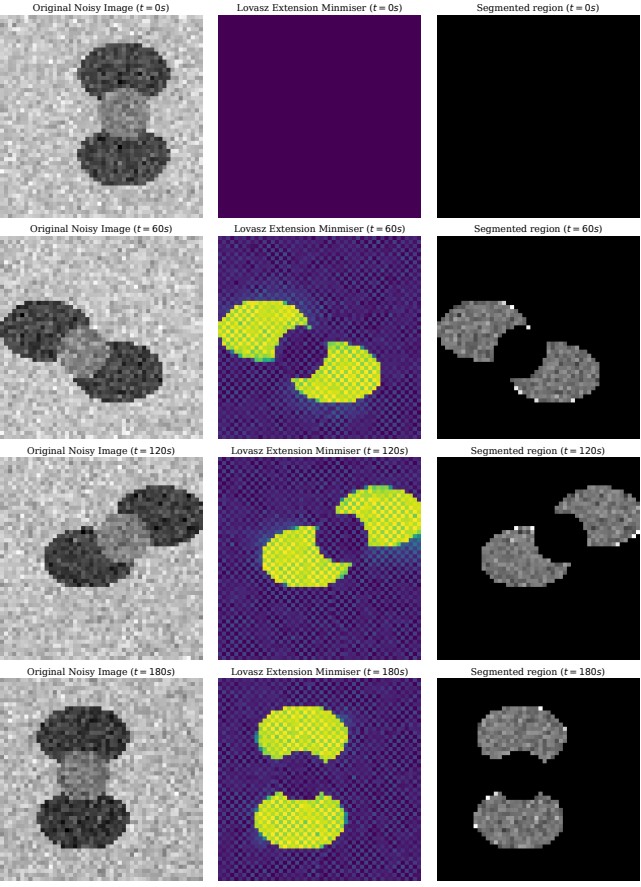

*Figure 5.* Online image segmentation under adversarial attack using Algorithm 1 at different times.

where $\rho \geq 0$, $\mathcal{Y}_\rho = \{ y \in [0,1]^{|M|} : \sum_i y_i \leq \rho \}$ constrains the attack budget, and the cost function $f(A, y)$ is defined as

$$f(A,y) = I(f_A, f_{V \setminus A}) - \sum_{k \in (V \setminus A) \cup U} \log(1 - \eta_k) - \sum_{k \in A \cup U} \log \eta_k - \sum_{k \in M} \left[ (1 - y_k) \log p_k^n + y_k \log p_k^f \right], \tag{68}$$

with $U = V \setminus M$ denoting the unlabelled indices, and

$$p_k^n = \eta_k^{\mathbb{I}[k \in A]} (1 - \eta_k)^{\mathbb{I}[k \notin A]}, \qquad p_k^f = (1 - \eta_k)^{\mathbb{I}[k \in A]} \eta_k^{\mathbb{I}[k \notin A]}.$$

The first term in (68) measures the mutual information between the Gaussian processes associated with the two clusters, while the remaining terms penalise deviations from the prior probabilities $\eta_k$ and the adversarial label flips encoded by $y$. Similar to the explanation given in (Bach, 2013), it can be seen that $f(A, y)$ is submodular with respect to $A$. Moreover, $f(A, y)$ is affine and thus concave with respect to $y$. We use Algorithm 1 to solve problem (67). First, we test the trivial cases when $\rho = \infty$ and $\rho = 0$. To have more uncertainty, we can impose $\eta_k \in \{0.1, 0.9\}$ for the labelled data and $\eta_k = \frac{1}{2}$ for the rest. We test the algorithm with $N = 1500$, $h_1 = h_2 = 10^{-3}$, and $\mu = 10^{-3}$. The results can be seen in Figure 7. As expected, when the attacker has an infinite budget, the clustering is completely corrupted and when the attacker has zero budget, the clustering is fully completed. Next, we solve the problem with the limited budget for the attacker and set $\rho = 2$. The result is shown in Figure 8. It can be seen that even when the attacker has enough budget to flip two of the labels completely, the clustering is completed successfully, and the semi-supervised clustering has been robust to the uncertainties. Next, we investigate the relation between the attacker budget and the clustering accuracy. We set $N = 2500$, $h_1 = h_2 = 10^{-4}$, $\mu = 10^{-6}$, and for $\rho \in [2, 5]$ we increase the budget, run the algorithm and measure the final clustering accuracy. The corresponding results can be seen in Figure 9.

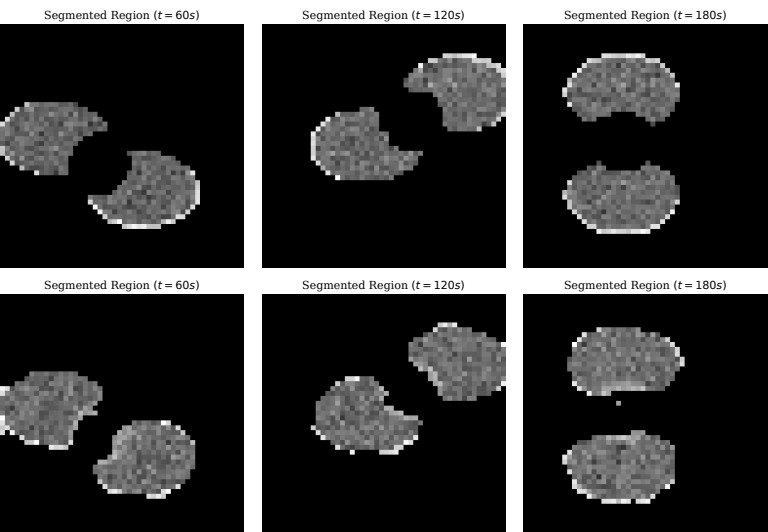

*Figure 6.* Online image segmentation at different times. Top row is predicted by the supervised U-Net and bottom row is predicted by the semi-supervised U-Net.

## E.4. Online Adversarial Attack on Semi-Supervised Clustering

In this section, we solve the online setting of the problem introduced in Section E.3, where each of the true clusters is moving with a different trajectory in each iteration. Thus, $K_{VV}$, defined in Section E.3, changes constantly, leading to an online problem. We solve the problem using Algorithm 1 and we set $N = 6000$, $h_1 = h_2 = 10^{-4}$, $\mu = 10^{-6}$ and $\rho = 2$. In this case, we consider 40 points and 8 randomly chosen labelled points. The Lovász extension value versus iterations is given in Figure 10. Moreover, the final clustering is given in Figure 11. As can be seen, Algorithm 1 has successfully completed the online clustering task under adversarial attacks. Moreover, the positions and predicted labels of data points over iterations can be found in Figure 12. Similarly to the offline case, one can see that with more budget, the attacker can disrupt the clustering completely.

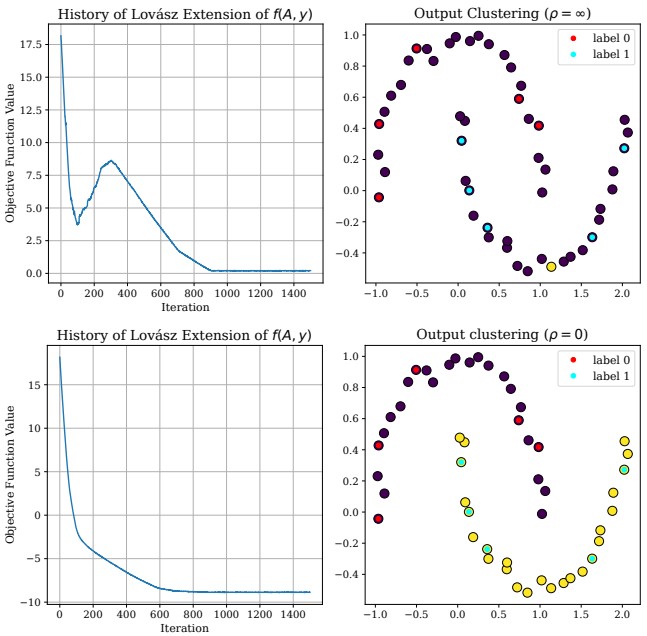

*Figure 7.* Lovász extension history over the sequence generated by Algorithm 1 and the offline clustering for $\rho = \infty$ and $\rho = 0$.

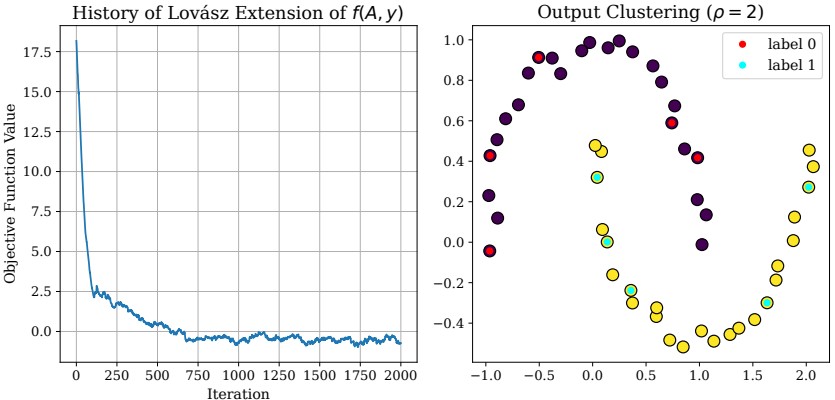

*Figure 8.* Lovász extension history over the sequence generated by Algorithm 1 and the offline clustering for $\rho = 2$.

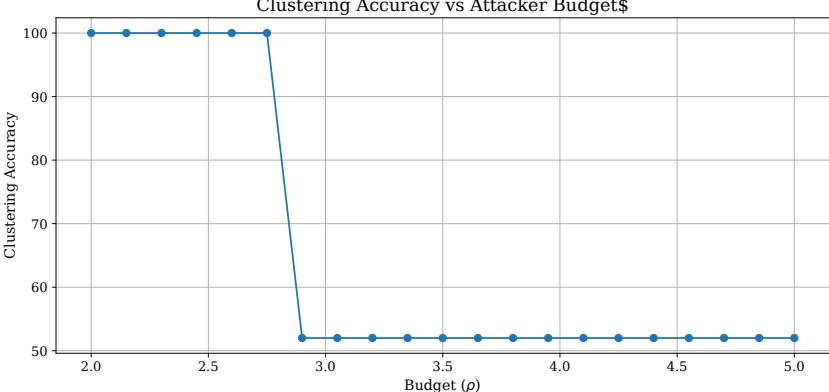

*Figure 9.* Offline clustering accuracy versus the attacker's budget.

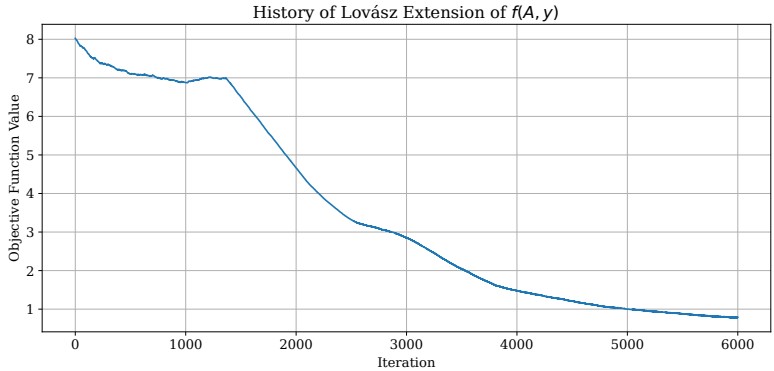

*Figure 10.* Lovász extension history over iterations for online clustering.

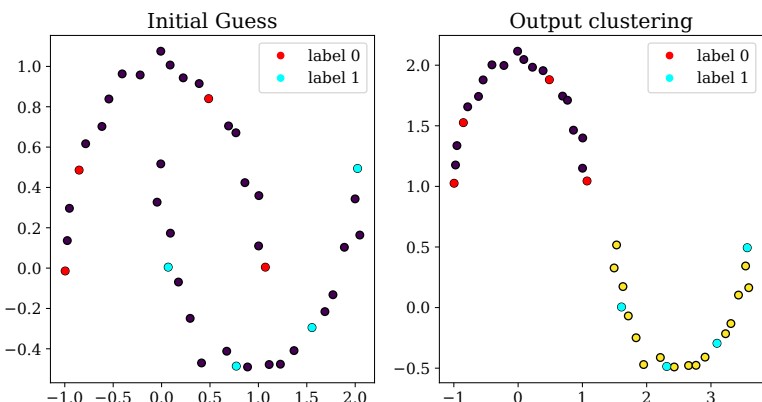

*Figure 11.* Online clustering of the Two Moon dataset under adversarial attack using Algorithm 1.

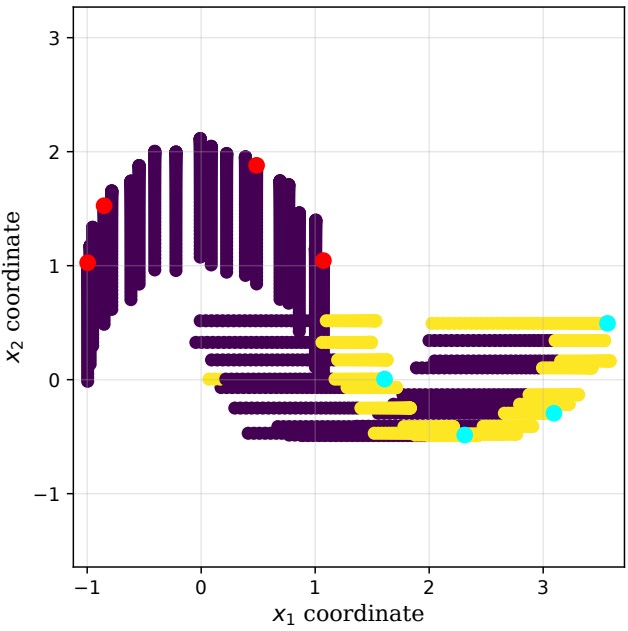

*Figure 12.* Positions and labels over iterations - online robust clustering.

