# OpenReview forum: "Solving the Offline and Online Min-Max Problem of Non-smooth Submodular-Concave Functions: A Zeroth-Order Approach"
_ICML.cc/2026/Conference — ICML 2026 regular_

### Official Review · Reviewer_MpXA · 2026-02-19

**Soundness:** 3
**Presentation:** 3
**Significance:** 3
**Originality:** 2
**Overall Recommendation:** 5
**Confidence:** 3

**Summary:**

This study focuses on non-smooth max-min/min-max optimization problems and proposes a zeroth-order optimization method. The approach combines the subgradient technique based on the Lovász extension with respect to the minimizer and Gaussian smoothing-based gradient estimation for the maximizer, enabling gradient-free optimization. Theoretically, the paper proves the convergence to an \epsilon-saddle point, while in the online setting, it achieves an upper bound on the duality gap related to the path length of the optimal decision sequence. Additionally, the study provides a complete complexity analysis and hyperparameter selection guidelines, supported by numerical experiments that validate the theoretical findings.

**Compliance With Llm Reviewing Policy:**

Affirmed.

**Final Justification:**

accept

**Key Questions For Authors:**

1.  page 4, equation (4) appears to be incorrect.
2. In Appendix E (Figure 3, Offline Image Segmentation), a surge in the cost function is observed. How should this phenomenon be explained?
2. In Figure 4, the objective function value does not show a clear decreasing trend. How is it ensured that the algorithm has reached a saddle point?

**Limitations:**

1. Dependence on specific problem structures:​ The effectiveness of the method heavily relies on the specific structure where the problem is "submodular with respect to the minimizer and concave with respect to the maximizer." The paper does not thoroughly explore how the algorithm's performance degrades when the problem slightly deviates from this structure (e.g., in cases of approximate submodularity or weak concavity), which limits its applicability to broader classes of mixed-integer or non-convex optimization problems.
2. Practical cost of hyperparameter selection:​ Although hyperparameter selection guidelines are provided, the optimization process itself may incur additional computational overhead (such as internal cross-validation). For large-scale or online learning scenarios, this tuning cost could offset the theoretical efficiency advantages of the algorithm, a point that has not been sufficiently discussed in the paper.

**Strengths And Weaknesses:**

Strength:
The paper addresses a theoretically and practically significant class of non-smooth max-min optimization problems with a submodular-concave structure. Its core innovation is a novel zeroth-order algorithm that elegantly combines the Lovász extension for the discrete minimizer and Gaussian smoothing for the continuous maximizer. The theoretical analysis is comprehensive, providing convergence guarantees for both offline and online settings, along with complexity results and hyperparameter guidelines. Numerical experiments offer preliminary validation of the theoretical findings.

Weaknesses:
The experimental section is primarily illustrative rather than comparative. The lack of systematic benchmarks against some other established optimization methods makes it difficult to assess the algorithm's relative performance, efficiency, and competitiveness. Furthermore, the presentation lacks in-depth analysis of convergence behavior, cost function fluctuations and hyperparameter sensitivity, which limits the persuasiveness and practical insight offered by the results.

---

> ### Author Rebuttal · Authors · 2026-03-29
>
> We thank reviewer "MpXA" for the precise summary and valuable comments. We are encouraged that the reviewer found the paper technically sound, and we address concerns mainly related to numerical examples below. We believe these concerns can be resolved through clarification and minor revisions.
>
> **Weaknesses:**
>
> 1. We thank the reviewer for the positive assessment and insightful suggestions. We acknowledge that additional benchmarks would strengthen the experimental section. We would like to note that, to the best of our knowledge, no prior optimisation method exists for the non-smooth submodular-concave min-max problem class, making selecting appropriate baselines challenging. The U-Net comparison in Table 1 is fairly thorough, reporting IoU, precision, recall, F1 score, memory footprint, and real-time capability across three methods. In the final version, we will add a more detailed discussion of convergence behaviour, including analysis of the variance across independent runs (which is on the order of $10^{-4}$ relative to reported values, as mentioned in Appendix E.2).
>
> **Key Questions:**
>
> 1. We thank the reviewer for pointing this out. We believe equation (4) on page 3 is correct as written: it defines the restricted gap $R(\bar{S}, \bar{y}) = f(\bar{S}, y^\ast) - f(S^\ast, \bar{y})$, measuring the sub-optimality of a candidate pair $(\bar{S}, \bar{y})$ relative to the saddle point $(S^\ast, y^\ast)$. If the reviewer has a specific concern, we would be happy to address it further.
>
> 2. The surge in the cost function during the transition period before convergence is an observed phenomenon in min-max optimisation. It arises because the extragradient method alternates between the minimiser pushing the cost down and the maximiser pushing it up, causing transient oscillations before the two players reach equilibrium. The choice of step size also influences the magnitude of these oscillations. Importantly, our convergence guarantee (Theorem 3.2) is for the average expected duality gap and the best generated candidate, not for monotonic decrease at every iteration—this is standard in the optimisation and ZO literature [1,2]. As shown in Figures 1 and 3, despite the transient surges, the algorithm ultimately produces correct segmentations that match the ground truth. We will clarify this more in the revised manuscript.
>
> 3. Figure 4 corresponds to the online setting, where the cost function $f_k$ changes at every iteration (each frame of the video introduces new edge weights due to the rotating and moving object), as well as the solution. A monotonically decreasing trend is not expected unless variations vanish, which is not the case here. The algorithm's goal is to track the changing optimum, not converge to a fixed point. Evidence of near-saddle-point solutions at each frame is provided by: (a) Table 1, where Algorithm 1 achieves average IoU of 0.975, precision of 0.986, and F1 of 0.987 across all frames; and (b) the visual results in the supplementary videos and Figures 2 and 5, which show that the segmentation consistently matches the ground truth despite the changing cost function and adversarial perturbations. We will clarify this in the revised manuscript.
>
> **Limitations:**
>
> 1. The reviewer raises a valid point. The theoretical results rely on the exact submodular-concave structure. For approximate submodularity, the subgradient (Definition 2.9) would include a bounded error proportional to the deviation from exact submodularity. Such errors could in principle be absorbed into the convergence analysis with modified rates, similar to how inexact oracle frameworks are handled in the convex optimisation literature. For weak concavity, the Gaussian smoothing treatment would need to account for the deviation from concavity. Obtaining precise convergence guarantees for these more general cases is an interesting direction for future work, and we will discuss this in the final manuscript.
>
> 2. We acknowledge the general difficulties of calculating or estimating some of the parameters used in theorems like Lipschitz constants. But, after estimation, the hyperparameters do not require cross-validation or grid search. As specified in Theorems 3.2 and 3.5, all hyperparameters ($h_1$, $h_2$, $\mu$) are given in closed form as explicit functions of problem parameters (Lipschitz constants $L_0$, $L_{0y}$, dimensions $n$ and $m$, desired accuracy $\epsilon$, and constraint set diameter). We think online parameter estimation is an interesting problem, which we thought was out of the scope of this work and can be a focus of future work
>
> We hope these clarifications address your concerns, and we welcome any further questions. We are happy to incorporate all suggested revisions in the camera-ready version.
>
> ---
>
> [1] Nesterov & Spokoiny, 2017. Random gradient-free minimization. Found. Comput. Math
>
> [2] Farzin et al., 2025. Min-max optimisation for nonconvex-nonconcave functions using a random ZO extragradient algorithm. TMLR

---

> > ### Author Rebuttal · Reviewer_MpXA · 2026-04-01
> >
> > I appreciate your responses and have no further questions at this time.

---

### Official Review · Reviewer_pJMp · 2026-03-07

**Soundness:** 3
**Presentation:** 2
**Significance:** 2
**Originality:** 2
**Overall Recommendation:** 3
**Confidence:** 3

**Summary:**

This article discusses the min-max and max-min optimization with non-smooth submodular-concave objectives, in both offline and online settings. The paper studies problems where the minimizer acts over subsets of a ground set and the maximizer acts over a continuous convex domain using a zeroth-order approach.

The main idea is to lift the set-valued minimization variable to the Lovász extension, so that the discrete problem can be handled through a convex continuous surrogate in the minimizer variable, while the maximizer variable is handled with Gaussian smoothing and a two-point zeroth-order estimator. The resulting method combines:
- a Lovász subgradient with respect to the minimizer,
- a Gaussian-smoothed zeroth-order estimator with respect to the maximizer,
- an extragradient-style update for the coupled saddle-point problem.

The paper proves, in expectation:
- convergence to an $\epsilon$-saddle point in the offline setting,
- an online duality gap of order depending on the number of iterations and the path-length of the optimal decisions.

The work also includes numerical illustrations, especially an adversarial image-segmentation example, where the proposed method is compared against supervised and semi-supervised U-Net baselines.

**Compliance With Llm Reviewing Policy:**

Affirmed.

**Key Questions For Authors:**

1. Can you elaborate on the novelty (first point in the weaknesses)?
2. Can you discuss the rates/bounds relative to the related works?
3. Is there an estimation about the computational complexity of Algorithm 1 (compared to other smooth/nonsmooth minimax approaches)

**Limitations:**

It is a theoretical work; no specific point needs to be discussed.

**Strengths And Weaknesses:**

Strengths
1. Interesting problem setting

The paper targets a fairly unusual and technically interesting setting: mixed discrete-continuous min-max optimization where the discrete component is submodular and the continuous component is concave but possibly non-smooth. This combination is nonstandard and potentially useful.

2. Non-trivial analysis

While the use of Lovász extension to convexify the minimizer and Gaussian smoothing approach to estimate the gradient is standard, the analysis is non-trivial.

3. Offline and online analysis in one framework

It is valuable that the paper treats both:
- offline saddle-point approximation,
- online dynamic performance via duality-gap bounds.

4. Numerical illustrations are reasonably concrete

The image-segmentation example is more developed than the toy examples often seen in optimization papers. Table 1 on page 8 reports IoU, precision, recall, F1, and memory footprint, and the proposed method performs reasonably in those experiments.


Weaknesses

1. Novelty relative to prior work is not fully convincing

The paper claims this is the first study of non-smooth submodular-concave min-max optimization, but several components are already standard:
- Lovász-extension-based submodular minimization,
- zeroth-order Gaussian smoothing,
- extragradient methods for saddle-point problems.

The contribution appears to be mainly a combination of known ingredients adapted to this setting. That may still be worthwhile, but the novelty should be positioned more carefully.

2. Rates seem relatively standard or weak

The paper would benefit from a more explicit comparison to the best known rates in related smooth/nonsmooth continuous minimax settings.

3. Presentation is hard to follow

The paper is mathematically dense, and several points need clearer exposition:

- the distinction between the duality gap and the restricted gap,
- The statements of Theorems 3.2 and 3.5 are long and may be stated in a more elegant manner

Other comments:

1. Provide a concrete example in the introduction for the setting. It is interesting to see that it is the game between a player who can choose only an integer solution (a subset) and another player who can choose a continuous solution. Provide concrete examples to motivate the setting?

---

> ### Author Rebuttal · Authors · 2026-03-29
>
> We thank reviewer "pJMp" for the valuable comments and suggestions. We are encouraged that the reviewer found the paper technically sound, and we address concerns mainly related to novelty, presentation and clarity below. We believe these concerns can be resolved through clarification and minor revisions.
>
> **Weaknesses:**
>
> 1. As the reviewer mentioned, our paper considers a new and technically interesting setting with non-trivial analysis.  While we leverage tools from prior works [4-7], our method and analysis for this setting have not been studied before. The novelty is threefold:
>
>    **(1) New problem class.** To the best of our knowledge, this is the first study addressing non-smooth submodular-concave min-max/max-min problems in both offline and online settings. Without accessing gradient information, such a setting is useful to applications such as adversarial black-box image segmentation. Prior work [1] considered convex-submodular objectives, and [2] studied purely combinatorial submodular min-max. Our work fills the mixed-integer gap where the minimiser is combinatorial (submodular) and the maximiser is continuous (concave), with possible non-smoothness in both.
>
>    **(2) Novel algorithmic framework.** We propose a hybrid ZO extragradient method combining Lovász extension subgradients for the discrete variable with Gaussian smoothing for the continuous variable, specifically tailored to the non-smooth structure on both sides and requiring no first-order information.
>
>    **(3) New theoretical results.** Theorem 3.2: offline convergence to an $\epsilon$-saddle point with $\mathcal{O}(nm^2\epsilon^{-2})$ queries, via novel analysis of the interplay between Lovász extension non-smoothness and Gaussian smoothing bias. Theorem 3.5: first $\mathcal{O}(\sqrt{N\bar{P}_N})$ online pseudo-duality gap bound for submodular-concave min-max, extending dynamic regret-style analysis to this mixed setting. Propositions 2.10–2.11: new sufficient conditions for saddle point existence (not analysed in [1]) and relations between (1) and (7). Additionally, the adversarial image segmentation application is new, outperforming supervised and semi-supervised U-Nets without pretraining, with adversarial robustness by design.
>
> 2. For continuous non-smooth convex-concave min-max, first-order subgradient methods achieve $\mathcal{O}(\epsilon^{-2})$ iteration complexity [3]. Our Theorem 3.2 matches this $\epsilon$-dependence but introduces: (i) $m^2$ from ZO Gaussian smoothing variance in $\mathbb{R}^m$, standard for ZO methods [4]; (ii) $n$ per-iteration cost from Lovász extension and subgradient computations over $[n]$. Total sample complexity: $\mathcal{O}(nm^2\epsilon^{-2}),$ standard using subgradients (8) [5]. The $\epsilon$-dependence matches the best known rates, and the additional factors are the natural cost of operating without gradients and handling combinatorial structure, consistent with information-theoretic costs of ZO and submodular components. Comparison with [1], is given in Weakness 4 of the response to the reviewer H2mL.
>
> 3. We appreciate the feedback, and we believe rigorous presentation of the theoretical results is important. We will improve the presentation: (i) add a remark after Definition 2.2 clarifying the relationship between duality gap $D(\bar{S}, \bar{y})$ and restricted gap $R(\bar{S}, \bar{y})$, noting that $0 \leq R \leq D$ where the restricted gap uses saddle point knowledge while the duality gap is saddle-point-free; (ii) restructure Theorems 3.2 and 3.5 to be easier to follow.
>
> 4. We will add a concrete motivating example in the introduction: adversarial semi-supervised image segmentation, where a learner selects pixels $A \subseteq V$ for a segment (discrete, submodular cost via graph cut) while an adversary controls trust weights $y \in [0,1]^{|S|}$ on seed labels (continuous, concave). The learner minimises segmentation cost while the adversary maximises it by discrediting seed labels, yielding a submodular-concave min-max problem of form (1). This, along with the adversarial clustering example (Appendix E.3), will be described in the introduction.
>
> **Key Questions:**
>
> 1. Please refer to Weakness 1 above.
> 2. Please refer to Weakness 2 above.
> 3. Please refer to Weakness 2 above.
>
> We hope these clarifications address your concerns, and we welcome any further questions. We are happy to incorporate all suggested revisions in the camera-ready version.
>
> ---
>
> [1] Adibi et al., 2022. Minimax optimisation: The case of convex-submodular. AISTATS.
>
> [2] Mualem et al., 2024. Submodular minimax optimization. AISTATS.
>
> [3] Nedić & Ozdaglar, 2009. Subgradient methods for saddle-point problems. JOTA.
>
> [4] Nesterov & Spokoiny, 2017. Random gradient-free minimization. Found. Comput. Math.
>
> [5] Hazan, 2016. Introduction to online convex optimization. Found. Trends Optim
>
> [6] Bach, 2011. Learning with submodular functions. arXiv:1111.6453
>
> [7] Lovász, 1983. Submodular functions and convexity. Springer

---

> > ### Author Rebuttal · Reviewer_pJMp · 2026-04-01
> >
> > I appreciate your responses and have no further questions at this time.

---

> > > ### Author Response · Authors · 2026-04-02
> > >
> > > We thank Reviewer pJMp for reading our rebuttal. We note that the acknowledgement selects option (c), indicating unresolved concerns that touch the core tenets of the work, yet the stated reason is "I appreciate your responses and have no further questions at this time." We would be grateful if the reviewer could clarify which specific concerns remain unresolved so that we may address them.

---

### Official Review · Reviewer_8VP2 · 2026-03-11

**Soundness:** 4
**Presentation:** 4
**Significance:** 1
**Originality:** 1
**Overall Recommendation:** 3
**Confidence:** 4

**Summary:**

In this paper they consider a min-max submodular-concave problem.
The main approach is to relax  the submodular function to the continuous domain using the classical Lovasz extension.
Based on prior work, they show how to compute  gradients for both the min and max players. This enables them to run extra-gradient method and effectively reduce the problem to min-max convex-concave games.
Under appropriate assumptions (e.g., existence of saddle point, the solution of the fractional problem being equal to the solution of the original problem) they show that the proposed algorithm converges to an $\epsilon$-saddle point.
Furthermore, they generalize the problem (and algorithm) to its natural online version.
Finally, the conduct experiments that validate the effectiveness of the proposed method.

**Compliance With Llm Reviewing Policy:**

Affirmed.

**Key Questions For Authors:**

1. How restive are the assumptions, and in particular proposition 2.10 and 2.11?
It seems a very strong assumption to relax the problem to the fractional domain x \in [0,1]^n and then assume that the optimal value of the fractional problem occurs on a corner of the space x \in \{0, 1\}^n.
Under this assumption the problem seems to be exactly the same as min-max convex-concave optimization.
Also, is this assumption realistic in your experiments?

2. What is exactly the oracle model? How many queries are we allowed to do? It seems that each (sub-)gradient computation requires n queries for the x-player and we need to compute twice per iteration of the algorithm. Would it be more natural to assume full-information feedback where the function and gradient is revealed?

3. What do you mean by smoothness? I think in optimization there are many meanings (e.g. L-smoothness). Also what do you mean in line 167 “As $f^L$ is non-smooth, ZO methods are good candidates to solve (7)”.

4. What is the novelty of this paper and what is new in your results and theorems (see also weaknesses)?

Feel free to comment on the weaknesses as well.

**Limitations:**

yes

**Strengths And Weaknesses:**

Strengths:
- Well written and easy to follow
- All results are sound and based on formal proofs
- Experiments support the theoretical analysis. Also, the proposed algorithm is better than the baseline (U-Net).

Weaknesses:
- Strong assumptions: existence of $\epsilon$ saddle point, solution of the fractional problem is on the corners of the feasible set, i.e., $x \in \{0, 1\}^n$

- The main result (Theorem 3.2) is for $max_y E[f(S, y)]$ rather than $E[max_y f(S, y)]$. That is, the y-player selects y against the distribution over sets and not the realized set. I think it is more natural for the y-player to select against the realized set. Please clarify if my understanding is correct.
Note that $E[max_y f(S, y)] \geq max_y E[f(S, y)]$, hence the setting I propose is more difficult that the one you solve.

- In the Definition 2.3 and Definition 2.4 I think it’s best to reference the original works that introduced the concepts and not Hazan & Kale, 2012
Maybe for submodular function there is no need for a reference.

- The paper is heavily based on prior work and mainly Hazan & Kale, 2012. For example, the use of the Lovasz Extension, gradient computation, and the conversion of the problem to convex optimization. Smoothing is based on Nesterov & Spokoiny, 2017 (Section 2.4).
Also, is Theorem 3.2 a new result? I expect that min-max convex-concave problems have already been solved. The same holds for Theorem 3.5 (which I think is based on Zhang et al, 2022).


nit: In line 326 I think you don’t mean expected regret but expected duality gap. Although if the expectation in inside the maximum I would call this pseudo-duality gap instead of expected duality gap similar to how we define pseudo- and expected regret.

---

> ### Author Rebuttal · Authors · 2026-03-29
>
> We thank reviewer "8VP2" for the valuable comments. We are encouraged that the reviewer found the paper technically sound. We address the concerns below.
>
> **Weaknesses:**
>
> 1. The existence of a saddle point is a standard assumption in min-max literature, analogous to assuming the objective is bounded below in minimisation. We clarify that we did not assume solutions lie on vertices of the feasible set. We only assumed that a solution to (1) (not (7)) exists (which by construction is on the vertices as one variable is discrete). We present sufficient conditions (Propositions 2.10–2.11) guaranteeing existence and revealing the relation between (1) and (7). We showed that solving the minimax via the Lovász extension is equivalent to solving the original maximin (proof of Proposition 2.10), which by the saddle point assumption for (1), is also a solution to the minimax version. Moreover, Proposition 2.10 does not require that all solutions are in $\\{0,1\\}^n$; it only requires that there exists one. These are sufficient conditions, not necessary ones.
>
> 2. Theorem 3.2 guarantees $\max_{y \in \mathcal{Y}} E_\tau[f(\hat{S}\_k, y)]$ rather than the realised $f(\hat{S}\_k, y)$. This is inherent to the Lovász extension: threshold rounding in (16) produces a distribution over sets, standard in submodular optimisation [2,4]. Since $E_\tau[f(S_\tau, y)] = f^L(x, y)$ (Lemma 2.7), so the guarantee is tight w.r.t. the Lovász extension. Obtaining guarantees for the realised set is an interesting direction for future work.
> 3. We will add the original reference.
>
> 4. Our work addresses a non-smooth submodular-concave min-max setting, new in the literature yet useful for applications such as adversarial image segmentation. While we leverage tools from prior works [2–4,6], our method and analysis for this setting have not been studied before. Theorems 3.2 and 3.5 are new and problem-fit. The novelty lies in analysing the interaction between Lovász extension properties [6], Gaussian smoothing [3], and the extragradient framework under ZO information. We clarify that Theorem 3.5 is *not* based on [5], which considered smooth bilinear games with a different structure and proof technique.
>
> 5. Noted. We will revise the wording.
>
> **Key Questions:**
>
> 1. As in Weakness 1, we only assume a solution for (1) exists (not(7)). Propositions 2.10–2.11 give sufficient (not necessary) conditions. Our numerical examples (code in supplementary) confirm the algorithm successfully solves the problems, indirectly validating existence.
>
> 2. Per iteration: 2 subgradients require $2n$ evaluations; 4 Lovász extensions for the ZO oracle require $4n$. Total: $6n$ per iteration. With $\mathcal{O}(m^2\epsilon^{-2})$ iterations (Theorem 3.2), total complexity is $\mathcal{O}(nm^2\epsilon^{-2})$. The ZO setting is useful when gradients are costly (e.g., black-box adversarial settings). We address the most restrictive setting.
>
> 3. By smoothness, we mean continuously differentiable; we will clarify. The Lovász extension is non-differentiable [2,4] and we allow non-differentiability w.r.t. the maximiser, making ZO methods a natural choice.
>
> 4. The novelty is threefold:
>
>    **(1) New problem class.** To our knowledge, this is the first study of non-smooth submodular-concave min-max in both offline and online settings. [1] considered convex-submodular objectives; [7] studied purely combinatorial submodular min-max. We fill the mixed-integer gap where the minimiser is combinatorial (submodular) and the maximiser is continuous (concave), with possible non-smoothness in both.
>
>    **(2) Novel algorithm.** A hybrid ZO extragradient combining Lovász extension subgradients (discrete variable) with Gaussian smoothing (continuous variable), requiring no first-order information.
>
>    **(3) New theory.** Theorem 3.2: offline convergence to $\epsilon$-saddle point with $\mathcal{O}(nm^2\epsilon^{-2})$ queries, via novel analysis of Lovász non-smoothness and Gaussian smoothing interaction. Theorem 3.5: $\mathcal{O}(\sqrt{N\bar{P}_N})$ online bound for non-smooth submodular-concave min-max. Prps 2.10–2.11: new sufficient conditions for saddle point existence (not analysed in [1]). The adversarial image segmentation application is also new, outperforming supervised and semi-supervised U-Nets without pretraining.
>
> We hope the clarifications address your concerns, and we welcome any further questions. We are happy to incorporate all suggested revisions in the final version.
>
> ---
>
> [1] Adibi et al., 2022. Minimax optimisation: The case of convex-submodular. AISTATS
>
> [2] Hazan, 2016. Introduction to online convex optimization. Found Trends Optim
>
> [3] Nesterov et al., 2017. Random gradient-free minimization. Found Comput. Math
>
> [4] Bach, 2011. Learning with submodular functions.
>
> [5] Zhang et al., 2022. No-regret learning in time-varying zero-sum games. ICML
>
> [6] Lovász, 1983. Submodular functions and convexity. Springer
>
> [7] Mualem et al., 2024. Submodular minimax optimization. AISTATS

---

> > ### Author Rebuttal · Reviewer_8VP2 · 2026-04-03
> >
> > Thank you for addressing my questions.
> >
> > I would like to follow-up regarding my novelty concern, so that I am sure I have the correct understanding.
> > My understanding is that after making the assumption described in Proposition 2.10, i.e.
> > $$
> > \min_{x\in[0,1]^n} \max_{y \in Y} f^L(x, y) =
> > \min_{x\in\\{0,1\\}^n} \max_{y \in Y} f^L(x, y)
> > $$
> > you have effectively reduced the problem to min-max convex-concave optimization (since the Lovasz extension is convex) and both optimization variables are over a continuous domain.
> > Then, solving min-max convex-concave problems is well studied.
> > This goes beyond just verifying existence of a saddle point to a regime where algorithms already exist.
> >
> > Hence, I claim that since minimization of submodular functions via the Lovasz extension is already known and min-max
> > convex-concave optimization admits well known algorithms (zeroth order extra gradient), the novelty is the composition of these techniques.
> > Please correct me if I am wrong.
> >
> > At this stage, I will keep my score.

---

> > > ### Author Response · Authors · 2026-04-04
> > >
> > > We thank the reviewer for the follow-up. We address the unresolved concern below.
> > >
> > > ### 1. Proposition 2.10 is a structural result, not a hidden assumption
> > >
> > > The reviewer suggests that the standard saddle-point assumption for (1) already gives the equality
> > > $$\min_{x \in [0,1]^n} \max_{y \in \mathcal{Y}} f^L(x,y) = \min_{x \in \\{0,1\\}^n} \max_{y \in \mathcal{Y}} f^L(x,y),$$
> > > and hence the reduction to (7) is "already given." This is not correct. The chain of reasoning is as follows:
> > >
> > > - Lemma 2.7 proves that $f^L(x,y)$ is jointly convex-concave. Convexity of the Lovász extension in $x$ alone is classical [R1], but convex-concavity of the joint function requires a separate non-trivial proof due to the interaction between $x$ and $y$.
> > > - Sion's minimax theorem then gives $\min_{x \in [0,1]^n} \max_{y \in \mathcal{Y}} f^L = \max_{y \in \mathcal{Y}} \min_{x \in [0,1]^n} f^L$, and the proof of Proposition 2.10 uses Lemma A.1 to further establish $\min_{x \in [0,1]^n} \max_{y \in \mathcal{Y}} f^L(x,y) = \max_{y \in \mathcal{Y}} \min_{S \subset Q]} f(S,y)$.
> > > - Only then does the standard saddle-point existence assumption for (1) ($\underset{y \in \mathcal{Y}}{\max}\underset{S \subset Q}{\min}\;f(S,y) = \underset{S \subset Q}{\min}\underset{y \in \mathcal{Y}}{\max}\;f(S,y)$), which is commanly assumed in the minimax literature [R2, R3], yield $\min_{S \subset Q} \max_{y \in \mathcal{Y}} f(S,y) = \min_{x \in [0,1]^n} \max_{y \in \mathcal{Y}} f^L(x,y),$ allowing us to solve (7) instead of (1).
> > >
> > > Moreover, Proposition 2.10 provides an if condition (not if and only if), and Section 2.3 just gives some explanations on the relation between (1) and (7), and imposes no additional assumptions on (1).
> > >
> > > ### 2. The Lovász extension is a tool; the optimisation setting is new
> > >
> > > The convexity of the Lovász extension has been known since [R1], and approximately 30 years later, works such as [R4] used it as a tool for submodular minimisation. We similarly use this established tool, but to address a new and previously unstudied setting: min-max problems with submodular-concave cost functions. The novelty lies in how this tool is deployed in a mixed-integer two-player setting, not in the tool itself.
> > >
> > > ### 3. The hybrid ZO extragradient approach is not well-studied
> > >
> > > The related prior work splits into two cases. Sadiev et al. [R6] study ZO methods for smooth convex-concave saddle points, a fundamentally easier setting where gradient approximation is far more reliable. Dvinskikh et al. [R7] study ZO methods for nonsmooth convex-concave saddle points, but their method uses random unit vectors (not Gaussian directions) applied symmetrically to both variables, is offline only, and has no extragradient structure. Our algorithm is a hybrid: it uses the Lovász subgradient for $x$ and a Gaussian ZO oracle [R8] for $y$ within an extragradient framework. The asymmetric design is deliberate and justified in Remark D.1 in [R5]: using a Gaussian ZO oracle for $x$ is at least twice as expensive as the Lovász subgradient, since computing the Lovász extension requires $n$ function evaluations. Thus, our algorithm is efficiently designed for this setting. Furthermore, to our knowledge, no prior work establishes a ZO online min-max duality gap bound of the form $O(\sqrt{N\bar{P}_N})$; online ZO methods with path-length dynamic regret exist only for minimisation [R5, R9], not for min-max problems. To the best of our knowledge, no prior work combines (i) Lovász subgradients, (ii) Gaussian zeroth-order oracles, and (iii) extragradient updates in a mixed discrete–continuous offline and online min-max setting of this form.
> > >
> > > ---
> > >
> > > [R1] L. Lovász. Submodular functions and convexity. *Mathematical Programming: The State of the Art*, 1983.
> > >
> > > [R2] A. Adibi, A. Mokhtari, and H. Hassani. Minimax optimization: The case of convex-submodular. *AISTATS*, 2022.
> > >
> > > [R3] L. R. Mualem et al. Submodular minimax optimization: Finding effective sets. *AISTATS*, 2024.
> > >
> > > [R4] E. Hazan and S. Kale. Online submodular minimization. *JMLR*, 13(10), 2012.
> > >
> > > [R5] A. A. Farzin et al. Minimisation of submodular functions using Gaussian zeroth-order random oracles. *arXiv:2510.15257*, 2025.
> > >
> > > [R6] A. Sadiev, A. Beznosikov, P. Dvurechensky, and A. Gasnikov. Zeroth-order algorithms for smooth saddle-point problems. *MOTOR 2021*, Springer, 2021.
> > >
> > > [R7] D. Dvinskikh, V. Tominin, I. Tominin, and A. Gasnikov. Noisy zeroth-order optimization for non-smooth saddle point problems. *MOTOR 2022*, Springer, 2022.
> > >
> > > [R8] Y. Nesterov and V. Spokoiny. Random gradient-free minimization of convex functions. *Foundations of Computational Mathematics*, 17(2), 2017.
> > >
> > > [R9] I. Shames, D. Selvaratnam, and J. H. Manton. Online optimization using zeroth order oracles. *IEEE Control Systems Letters*, 4(1), 2019.

---

### Official Review · Reviewer_H2mL · 2026-03-15

**Soundness:** 3
**Presentation:** 2
**Significance:** 2
**Originality:** 2
**Overall Recommendation:** 4
**Confidence:** 2

**Summary:**

This work considers a class of minimax-problem with non-smooth and submodular-concave objective functions f(x,y). The applied the Lovasz extension to the variable x and extend the discrete optimiation w.r.t. x to a continuous optimisation over the unit hypercube. The extension converts the non-smooth submodular-concave to a non-smooth convex-concave problem. They characterise the existence of saddle point for the original problem. Authors consider the cases when only zeroth-order information is accessible and use a two-point estimator for gradient estimation. Then, convergence result (in expectation) are provided for both offline and online settings. Simulations are conducted using adversarial image segmentation using 50x50 grayscale images (with no codes provided).

**Compliance With Llm Reviewing Policy:**

Affirmed.

**Final Justification:**

The rebuttal has addressed most of my concerns.

**Key Questions For Authors:**

Please refer to the weaknesses.

**Limitations:**

yes

**Strengths And Weaknesses:**

### Strengths

* The paper is nicely written and easy to follow. Authors are familiar with related literature and provide a nice literature review for readers.

* The authors are rigorous with the assumptions and mathematical presentation.

### Weaknesses

* Writing: It would be helpful to organise the pseudo-algorithm in blocks and add comments for each blocks to explain the purpose of each block.

* It would be helpful to provide a link to the codes for simulations for completeness.

* It would be helpful to explain the availability of two-point estimator in practice since it reduces the variance by requiring more queries of function values compared to one-point estimator.

* In the related work section, one work working on the off-line convex-submodular minimax is cited, it would be helpful to have a comparison between this work and results in the offline settings.

* What is the dependence of iteration number w.r.t. dimension $n$ since ZO estimator is used? What would be the sample complexity in total? Even though the authors discuss ZO estimator in the main text, only results in expectation are provided. Since ZO two-point estimator is introduced in the main text, it would be very helpful to provide the total complexity and compare with the prior work such as (Adibi et al., 2022).

* In the main result (Theorem 3.2), dependence over $\bar{r}_0=\max_{z\in\mathcal{Z}}\|z_0-z\|$, m, and other parameters could benefit from further explanation, and it would also be helpful for readers to understand the contribution of this work by comparing with prior complexity (e.g. for continuous non-smooth convex-concave minimax problem).

* In the online setting, the dependence over $\bar{P}_N$ could use more explanation, why this term is needed for the result and when will this regret bound be meaningful, i.e., $\bar{P}_N<<O(N)$?

---

> ### Author Rebuttal · Authors · 2026-03-29
>
> We thank reviewer "H2mL" for the valuable comments and suggestions. We are encouraged that the reviewer found the paper technically sound. We believe the concerns, mainly related to clarity, can be resolved through minor revisions.
>
> **Weaknesses:**
>
> 1. In the final version, we restructure the algorithm into three clearly labelled blocks: (i) forward step (Lines 3–7), (ii) extrapolation step (Lines 8–13), and (iii) projection and update (Line 14), each with inline comments explaining its purpose.
>
> 2. Per ICML anonymity guidelines, all codes (along with 4 videos of online image segmentation) are in the supplementary zip file. A public GitHub link will be added in the final version.
>
> 3. The two-point estimator incurs only a marginal additional cost in practice (one additional function query): sampling $t$ directions, the one-point estimator needs $t$ queries while the two-point needs $t+1$, since $f(y)$ is reused across all directions. The order of function calls is unchanged while variance is reduced, making it standard in the ZO literature [5]. In Remark D.2 (Appendix D), we also discuss that the central difference estimator yields the same theoretical bounds in the offline case.
>
> 4. The key differences with [1] are: (a) [1] considers a convex-submodular min-max problem with a submodular *maximiser*, while we consider a submodular-concave min-max problem with a submodular *minimiser*. Since minimisation and maximisation of submodular functions are fundamentally different [2], the same techniques do not apply. (b) The convergence analyses differ entirely: [1] converges to an $(\alpha{-}\epsilon)$-approximate minimax solution (designed for submodular maximisation), while we converge to an $\epsilon$-saddle point. Both require $\mathcal{O}(\epsilon^{-2})$ iterations to converge to their goal. (c) [1] assumes gradient access, whereas our method only requires function evaluations.
>
> 5. Per iteration cost of Algorithm 1: (a) computing 2 subgradients (Definition 2.9) requires $2n$ evaluations of $f$; (b) evaluating 4 Lovász extensions for the ZO oracle (see (14)) requires $4n$ evaluations. Total per iteration: $6n$ evaluations. With $N = \mathcal{O}(m^2\epsilon^{-2})$ iterations (Theorem 3.2), the total sample complexity is $\mathcal{O}(n \cdot m^2 \epsilon^{-2})$. The ZO estimator acts only on $y \in \mathbb{R}^m$, so iteration count depends on $m$ not $n$; the factor $n$ is purely per-iteration cost from the Lovász extension and subgradient computations. We note that it is standard in the ZO optimisation literature [5] that all results are stated in the expectation sense for the randomness arising in ZO sampling. Direct comparison with [1] is not applicable due to fundamentally distinct problem settings (see Point 4).
>
> 6. For non-smooth convex-concave min-max problems, first-order subgradient methods achieve $\mathcal{O}(\epsilon^{-2})$ iteration complexity [6]. Our Theorem 3.2 matches this $\epsilon$-dependence but introduces: (i) $m^2$ from ZO Gaussian smoothing variance in $\mathbb{R}^m$, standard for ZO methods [5]; (ii) $n$ per-iteration cost from Lovász extension and subgradient computations over $[n]$ [2]. The $\bar{r}_0$ dependence is the constraint set diameter, standard in constrained optimisation bounds [3]. Our primary contribution is addressing the submodular-concave min-max problem class via ZO methods, not improving rates for the continuous convex-concave setting. To the best of our knowledge, this is the first study for this problem class. We will add this comparison to the manuscript.
>
> 7. The path length $\bar{P}_N$ captures how much optimal decisions change as cost functions evolve, standard in online optimisation [3,4]. The bound is $\mathcal{O}(\sqrt{N\bar{P}_N})$, so the average duality gap is $\mathcal{O}(\sqrt{\bar{P}_N / N})$, converging to zero whenever $\bar{P}_N = o(N)$. Concretely: (i) $\bar{P}_N = \mathcal{O}(1)$: gap is $\mathcal{O}(N^{-1/2})$, recovering the standard rate. (ii) $\bar{P}_N = \mathcal{O}(\sqrt{N})$: gap is $\mathcal{O}(N^{-1/4})$, still converging. (iii) $\bar{P}_N = \mathcal{O}(N)$: gap is $\mathcal{O}(1)$, no convergence guaranteed, as expected in highly varying and non-stationary regimes. This path-length dependence is inherent to the problem and consistent with the well-established behaviour of dynamic regret bounds in online optimisation [3].
>
> We hope the clarifications address your concerns, and we welcome any further questions. We are happy to incorporate all suggested revisions in the camera-ready version.
>
> ---
>
> [1] Adibi et al., 2022. Minimax optimisation: The case of convex-submodular. AISTATS
>
> [2] Bach, 2011. Learning with submodular functions. arXiv:1111.6453
>
> [3] Hazan, 2016. Introduction to online convex optimization. Found. Trends Optim
>
> [4] Zinkevich, 2003. Online convex programming. ICML
>
> [5] Nesterov & Spokoiny, 2017. Random gradient-free minimization. Found. Comput. Math
>
> [6] Nedić & Ozdaglar, 2009. Subgradient methods for saddle-point problems. JOTA

---

> > ### Author Rebuttal · Reviewer_H2mL · 2026-04-04
> >
> > I thank the authors for the thoughtful reply and decide to increase the score.

---

### Decision · Program_Chairs · 2026-04-30

**Decision:**

Accept (regular)

**Comment:**

The paper studies submodular-non smooth concave optimization. The new algorithm uses the Lovasz extension to extend the submodular function to a convex function in a continuous domain and then uses machinery for non-smooth convex-concave optimization to solve the problem. The reviewers are split on the paper with two recommending accept and two recommending reject. The paper considers a novel problem setting with reasonable motivation. However, the theoretical results require strong assumptions and the special cases where the assumptions are justified are limited (the only known ones in the paper are when the minimizer of a certain continuous function based on the objective is coincidentally discrete or when the min-max problem is essentially just min and the max argument is irrelevant). The reviewers are also concerned that there is no novel techniques but rather a combination of tools of several previous works. The combination is not trivial but at the same time lacking new ideas. The online results are not truly online but require additional assumption on having the same online function appearing multiple times in a row. I recommend weak accept.